# Response of the AMOC to reduced solar radiation – the modulating role of atmospheric-chemistry

Stefan Muthers[1,2,*], Christoph C. Raible[1,2], Eugene Rozanov[3,4], and Thomas F. Stocker[1,2]

[1]Climate and Environmental Physics, University of Bern, Bern, Switzerland
[2]Oeschger Centre for Climate Change Research, University of Bern, Bern, Switzerland
[3]Institute for Atmospheric and Climate Science, ETH, Zurich, Switzerland
[4]Physikalisch-Meteorologisches Observatorium Davos and World Radiation Center (PMOD/WRC), Davos, Switzerland
[*]now at: German Meteorological Service, Research Center Human Biometeorology, Freiburg, Germany.

*Correspondence to:* S. Muthers (muthers@climate.unibe.ch)

**Abstract.** The influence of reduced solar forcing (grand solar minimum or geoengineering scenarios like solar radiation management) on the Atlantic meridional overturning circulation (AMOC) is assessed in an ensemble of atmosphere-ocean-chemistry-climate model simulations. Ensemble sensitivity simulations are performed with and without interactive chemistry. In both experiments the AMOC is intensified in the course of the solar radiation reduction, which is attributed to the thermal effect of the solar forcing: reduced sea surface temperatures and enhanced sea ice formation increase the density of the upper ocean in the North Atlantic and intensify the deepwater formation. Furthermore, a second, dynamical effect on the AMOC is identified driven by the stratospheric cooling in response to the reduced solar forcing. The cooling is strongest in the tropics and leads to a weakening of the Northern polar vortex. By stratosphere-troposphere interactions, the stratospheric circulation anomalies induce a negative phase of the Arctic Oscillation in the troposphere which is found to weaken the AMOC through wind stress and heat flux anomalies in the North Atlantic. The dynamic mechanism is present in both ensemble experiments. In the experiment with interactive chemistry, however, it is strongly amplified by stratospheric ozone changes. In the coupled system, both effects counteract and weaken the response of the AMOC to the solar forcing reduction. Neglecting chemistry-climate interactions in model simulations may therefore lead to an overestimation of the AMOC response to solar forcing.

## 1 Introduction

The Atlantic meridional overturning circulation (AMOC) is an important component of climate variability in the North Atlantic region (Kuhlbrodt et al., 2007; Stocker, 2013). The surface branch of the AMOC transports heat from the Southern Hemisphere (SH) and the tropics towards the North, is closely connected to the Atlantic Multidecadal Oscillation, and contributes to the temperate climatic conditions in western Europe (Knight et al., 2006). Thus, understanding the variations in the strength of this circulation is important in particular for future climate change (Stocker and Schmittner, 1997; Manabe and Stouffer, 1999; Mikolajewicz and Voss, 2000; Gregory et al., 2005), decadal climate predictions (Griffies and Bryan, 1997; Meehl et al., 2009) and with respect to potential abrupt climatic changes as proposed for the past (Stocker and Wright, 1991; Stocker, 2000; Clark et al., 2002).

Several processes are involved in driving the AMOC ranging from internal processes of the climate system such as the thermohaline process (Wunsch, 2002; Kuhlbrodt et al., 2007; Lozier, 2010) to external forcing (Otterå et al., 2010). The purpose of this study is to assess the influence of a reduction of the solar forcing on the AMOC. In particular, we investigate the role of chemistry-climate interactions in modulating the response of the atmospheric circulation to reduced solar radiation and their effect on the AMOC. To this end, we perform ensemble sensitivity simulations for different solar radiation reductions with a state-of-the-art coupled atmosphere-ocean-chemistry-climate model, where atmospheric chemistry is either enabled or disabled.

So far, the external forcing response of the AMOC has been mainly studied in climate models without interactive atmospheric chemistry (Otterå et al., 2010). Thereby, volcanic eruptions have been found to intensify the AMOC on decadal time scales (Otterå et al., 2010; Mignot et al., 2011), through a reduction of sea surface temperatures and a shift of the North Atlantic Oscillation (NAO) towards its positive phase. Moreover, volcanic eruptions may excite the variability of the AMOC (Swingedouw et al., 2015). The response, however, may depend on the background conditions of the climate system (Zanchettin et al., 2012). An increase in the solar forcing has been found to weaken the AMOC by increasing SSTs and enhancing freshwater input (Cubasch et al., 1997; Latif et al., 2009; Otterå et al., 2010; Swingedouw et al., 2011) and has been proposed to be a driver of Greenland temperature variations (Waple et al., 2002; Kobashi et al., 2015).

Among many possible influences, a change in the solar forcing may also affect the NAO (Kodera, 2003; Ineson et al., 2011; Swingedouw et al., 2011; Scaife et al., 2013). For example, the circulation in the polar stratosphere during winter (polar night jet) has been proposed to be affected by a change in the ultra-violet (UV) radiation (Kodera and Kuroda, 2002). By stratosphere-troposphere interactions, stratospheric anomalies can propagate down to the troposphere and cause circulation anomalies at the surface (Perlwitz and Graf, 1995; Muthers et al., 2014a). A positive phase of the NAO is then associated with a strengthening of the polar night jet and vice versa (Baldwin and Dunkerton, 1999, 2001; Thompson and Wallace, 2001) and may also affect the AMOC (Manzini et al., 2012; Reichler et al., 2012).

The stratospheric response to UV variations is modulated by chemistry-climate interactions (Haigh, 1994, 1996). In particular, stratospheric ozone reacts to solar irradiance changes. The increase of solar UV enhances shortwave heating rate by the ozone absorption (e.g. Forster et al., 2011). Additional solar UV in the Herzberg continuum ($\lambda <$242 nm) intensifies ozone production, while UV in the Hartley band destroys ozone (e.g. Ball et al., 2016, Figure 1). Because the solar UV variability decreases with wavelength the first effect prevails and leads to ozone increase in the middle stratosphere in phase with the increase of the solar UV. In turn the ozone increase gives additional heating with magnitude comparable to primary heating by the increase of solar UV alone (Forster et al., 2011). This process can amplify the efficiency of the earlier mentioned top-down propagation (Kodera and Kuroda, 2002) and is obviously missing if the ozone concentration is prescribed.

Still, most of these studies are based on models without interactive atmospheric chemistry. The influence of climate changes on the state of the ozone layer has been recognized already long time ago. The cooling of the stratosphere by greenhouse gases (GHG) slows down catalytic ozone oxidation cycles leading to ozone increase (e.g. Haigh and Pyle, 1982; Revell et al., 2012). The greenhouse warming accelerates Brewer Dobson circulation reducing ozone in the tropical lower stratosphere and enhancing its abundance over middle to high latitudes (Deckert and Dameris, 2008; Zubov et al., 2013). The ozone changes

have substantial implications for the climate. The influence of the ozone recovery associated with the implementation of the Montreal Protocol limitations on the production of ozone destroying substances on the SH has been identified in the observations and model simulations (e.g. Son et al., 2008; Robinson and Erickson, 2015). Recently, it was suggested that the use of interactive chemistry instead of prescribed ozone climatology can influence climate model properties. Dietmüller et al.
(2014) showed that the application of interactive chemistry reduces the climate sensitivity by 3-8%. Similar reduction of the climate sensitivity was also found by Muthers et al. (2014b). A more substantial reduction of the model response to 4xCO2 by up to 20% due to taking into account interactive chemistry was reported by Nowack et al. (2014). All these studies attributed the reduction to the changes of ozone, water vapour and clouds in response to climate warming. These conclusions were not confirmed by very recent results of CESM1-WACCM model (Marsh et al., 2016), which found similar ozone response to
4xCO2 but no changes in climate sensitivity. In contrast, Chiodo and Polvani (2016) applied the same model and demonstrated that the interactive ozone introduces a negative feedback leading to a weaker surface warming due to an enhancement of the solar irradiance. Thus, these results show that further experiments are necessary in order to assess the model discrepancies and to deepen our understanding of the ozone feedback and its importance for the simulation of future climate change under the influence of different natural and anthropogenic factors.

The outline of this study is as follows. The model configuration and the experiments are described in section 2. Section 3 presents the results, first for the experiments without interactive atmospheric chemistry followed by an analysis of the differences causes by the chemistry-climate interactions. A summary and concluding discussion is given in section 4.

## 2    Model and experiments

### 2.1    The model

We use the coupled atmosphere-ocean-chemistry model SOCOL-MPIOM to simulate the effect of a change in the solar activity on the climate (Muthers et al., 2014b). SOCOL (Stenke et al., 2013) consists of the atmospheric component ECHAM5 (Roeckner et al., 2003) coupled to the chemistry module MEZON (Rozanov et al., 1999; Egorova et al., 2003). The middle atmospheric configuration of ECHAM5 is used (Manzini et al., 2006), which resolves the atmosphere up to $0.01\,\mathrm{hPa}$ (about $80\,\mathrm{km}$) with 39 levels. The horizontal resolution is T31, corresponding to a grid size of $3.75° \times 3.75°$.

The chemistry is directly coupled to ECHAM5 and uses temperature data to calculate the tendency of 41 gas species, taking into account 200 gas-phase, 16 heterogeneous, and 35 photolytical reactions. Optionally, the coupling to MEZON can be disabled. In this case a 3-dimensional time-dependent ozone data set needs to be specified.

    The short-wave radiation scheme of SOCOL considers spectral solar irradiance (SSI) values in six spectral bands. Time series for each spectral interval are used as forcing to allow for changes in the spectral composition of the total solar irradiance.
The short-wave scheme considers Rayleigh scattering, scattering on aerosols and clouds, and the absorption of UV by $O_2$, $O_3$, and 44 other species. Additional parametrizations for the absorption of UV in the Lyman-alpha, Schumann-Runge, Hartley, and Higgins bands are implemented following Egorova et al. (2004); Sukhodolov et al. (2014). The long-wave scheme considers

wavenumbers between $10\,\mathrm{cm}^{-1}$ to $3000\,\mathrm{cm}^{-1}$ and takes into account water vapour, $CO_2$, $O_3$, $N_2O$, $CH_4$, CFC-11, CFC-12, CFC-22, aerosols, and clouds.

With the given vertical resolution, SOCOL is not able to produce a Quasi-Biennial Oscillation (QBO). Thus a QBO nudging is applied (Giorgetta et al., 1999). The time-step of the atmospheric component is 15 minutes, with the full radiation and the chemical computations updates are performed every 2 hours.

SOCOL is coupled to the ocean model MPIOM (Marsland, 2003; Jungclaus et al., 2006) using the OASIS coupler (Budich et al., 2010; Valcke, 2013). MPIOM includes an embedded sea-ice module. To avoid numerical singularities at the North pole, both poles of the rotated Arakawa C grid are shifted and placed over land (Greenland and central Antarctica). The nominal resolution is $3°$ – varying between 22 and $350\,\mathrm{km}$ – with a higher resolution in the deep water formation regions in the North Atlantic and the Weddell Sea. Convection is implemented by greatly enhanced vertical diffusion, when the water column becomes unstable. Sea ice dynamics are based on the viscous-plastic rheology formulated by Hibler (1979). A constant sea ice salinity of 5 psu is assumed. The time-step of the oceanic component is 2 hours and 24 minutes.

## 2.2 The experiments

Ensemble sensitivity simulations with SOCOL-MPIOM are performed to study the effect of solar radiation reduction (SRR) on the climate system and the AMOC. Such SRRs are caused by either a grand solar minimum or solar radiation management techniques. 10 simulations are carried out for each ensemble experiment; the experiments differ in the solar forcing applied and whether or not chemistry-climate interactions are considered in the model.

Perpetual 1600 AD conditions and zero volcanic aerosols (i.e., excluding the volcanic eruption of Huaynaputina) are applied in all simulations. For the sensitivity simulations only the solar forcing is allowed to change in time. The solar forcing consists of the SSI and photolysis rates.

As reference experiment we perform two control ensembles, CTRL_CHEM and CTRL_NOCHEM, with and without interactive chemistry, respectively. In these experiments all forcings represent the conditions of the year 1600 AD, including the solar forcings of the year 1600 AD. The year 1600 was chosen, since a stable long-term control simulation with SOCOL-MPIOM was available from previous studies (Anet et al., 2013a, 2014; Muthers et al., 2014b). Note, the differences in the climatic conditions between 1600 and the commonly used year 1850 are small and both represent a preindustrial climate state.

The two SRRs simulated for this study are characterized by a step-wise total solar irradiance (TSI) reduction of $-3.5\,\mathrm{Wm}^{-2}$ and $-20\,\mathrm{Wm}^{-2}$, referred to as S1 and S2, respectively (Fig. 1a). The S1 SRR is comparable to a grand solar minima like the Dalton Minimum or Maunder Minimum in a large-amplitude solar forcing reconstruction (e.g. Shapiro et al., 2011). With $-20\,\mathrm{Wm}^{-2}$ the S2 SRR is comparable to a weak solar radiation management scenario (Kravitz et al., 2011), which may counteract an increase in the radiative forcing from GHG of about $3\,\mathrm{Wm}^{-2}$. The reduction of the solar forcings is switched on at year 5 of a simulation and lasts for 30 years when it is switched off and the simulation is continued for 25 years. Both SRRs are simulated with and without interactive chemistry and are named S1_CHEM, S2_CHEM, S1_NOCHEM, and S2_NOCHEM in the following. A summary of the experiments performed for this study is given in Table 1.

In the CHEM experiments, ECHAM5 and MEZON are coupled and the atmospheric chemistry responds to the solar radiation changes. In NOCHEM, temporal and spatial ozone variations need to be prescribed. Therefore, a daily 3D ozone climatology is applied, based on a 1600 AD control simulations.

All ensemble simulations are initialized from model year 1300 of a long control simulation with interactive chemistry performed under perpetual 1600 AD conditions (Muthers et al., 2014b). The ensemble members only differ in their initial conditions by slightly perturbing the atmosphere (atmospheric restarts for Jan-1, Jan-2, Jan-3, ...). The oceanic component is always initialized using the same initial conditions.

Note, that we erroneously applied a slightly different solar forcing in 6 of 10 simulations. This TSI difference of $0.018 \, \mathrm{Wm}^{-2}$ is caused by a different rounding of the SSI values and lead to very small differences between the control ensemble experiments and the SRR experiments, already prior to the start of the reduction.

The AMOC index is calculated by selecting the maximum in the annual mean meridional overturning streamfunction northward of 28°N and below 300 m. To detect the influence of the stratospheric circulation on the troposphere and the AMOC we use the hemispheric mode of the Northern Hemisphere (NH) the Arctic Oscillation (AO). While the NAO is more closely related to the AMOC, the AO has a stronger imprint of stratosphere troposphere interactions. The AO index is defined as the spatially averaged monthly mean sea level pressure difference between 40°N and 65°N, which is normalized by the mean and the standard deviation of the corresponding control ensemble. Furthermore, the index is multiplied by –1 to reflect the negative phase of the AO by negative values and vice versa. Using a different definition of the AO (EOF based or using the sea level pressure north of 70°N) or an index of the NAO leads to very similar results.

## 3    Results

Both SRRs leads to a significant reduction of the global mean near surface (2-m) air temperature (Fig. 1b). For the stronger S2 experiment the cooling is more pronounced than for the S1 experiments and reaches -1.0 K and -0.9 K for S2_CHEM and S2_NOCHEM, respectively (averaged over the last 5-yrs of the SRR period). For the S1 experiment, the temperatures reduces by -0.1 K in both ensembles. The temperature instantaneously responds to the imposed radiation drop and reaches the lowest values at the end of the reduction period. The continuous cooling in the course of the SRR, which is well visible in the S2 ensembles, suggests that the model has not yet reached thermal equilibrium. In fact, from the model's equilibrium climate sensitivity (for a doubling of $CO_2$, compare Muthers et al. (2014b)) an equilibrium temperature response of -1.3 K is expected for S2_CHEM and -1.4 K for S2_NOCHEM. However, a comparison with the $CO_2$ sensitivity is only a rough estimate, since the climate sensitivity (and the contributions from chemistry-climate interactions) differs between the solar and $CO_2$ forcing and depends on the sign of the forcing perturbation (Hansen et al., 1997; Schaller et al., 2014).

The larger cooling in the CHEM experiments is related to differences in the stratospheric response. In particular, stratospheric ozone concentrations are reduced due to the reduced UV radiation (Fig. S1 a and d), a process which is not considered in the NOCHEM experiments. Additionally, water vapour concentrations are affected by the SRR. In S2_NOCHEM, the largest anomalies (-15 %) are found in the tropical upper troposphere, but stratospheric reductions exceed -10 % almost everywhere

(Fig. S1c). In S2_CHEM, the stratospheric reductions in water vapour are more pronounced (up to -35 %), due to the effect of the solar forcing on the oxidation of methane, the most important in-situ source of stratospheric water vapour (Fig. S1b). Due to the greenhouse effect of ozone and water vapour, the outgoing long-wave flux increases more in CHEM than in the NOCHEM and leads to an additional cooling of the troposphere. The positive water vapour anomalies found in the uppermost model levels in the CHEM experiments (Fig. S1b and e) are related to the reduced UV photolysis of the water vapour molecules.

A slight initial reduction of the global mean temperature is also found in the reference ensemble experiments and is related to the initial conditions of the ocean. With all ensemble simulations sharing the same oceanic conditions in the beginning, the AMOC development of the first years is dominated by the oceanic memory. During the first decade of the experiments a decline of the AMOC from 21.0 to 19.8 Sv is found (Fig. 1c). This decline is very similar in both reference experiments. The minimum state of the AMOC is reached in year 12-13 of the reference experiments and in the following years the AMOC increases to its maximum value of 21.4 Sv in the year 35.

The AMOC is not affected by the SRR during the first few years of the simulation. Starting with simulation year 10, however, and even more pronounced in the second half of the reduction period, the AMOC is significantly stronger in S1_NOCHEM during several years and in S2_NOCHEM for most of the years between year 15 and 35 of the experiment. In the CHEM ensemble simulations no significant AMOC intensification is found for S1. In S2_CHEM, the AMOC is significantly stronger during the second half of the SRR period, but the intensification is weaker in comparison to S2_NOCHEM. The differences between the AMOC index for S2_CHEM and S2_NOCHEM are also reflected in the anomaly pattern of the AMOC (Fig. S2). Within the first 10 years the intensification of the circulation is weak. Positive anomalies are found between 40°N to 65°N and between the surface and a depth of 2800 m depth. During these first 20 years of the reduction period the intensification is slightly larger in S2_CHEM. A pronounced strengthening of the circulation occurs in the second decade of the reduction period. Positive anomalies cover all latitudes from the equator to 65°N and most levels between the surface and 3000 m depth. In the second decade the intensification is more pronounced in S2_NOCHEM. In the third decade, finally, a further intensification is found, which is again stronger in S2_NOCHEM. In the following, we will first address the relevant processes being responsible for the AMOC intensification (Sec. 3.1) before we assess the role of chemistry-climate interactions, to explain the lower sensitivity of the AMOC to SRR in the CHEM experiments (Sec. 3.2).

## 3.1 The thermal effect of SRR on the AMOC

A direct effect of SRR is the reduction in short-wave energy reaching the troposphere and the surface and thus in temperature, which is apparent almost everywhere in the NH (Fig. 2). Averaged over the 30-yr reduction period the sea ice growth in the Barents Sea is stronger in S2_CHEM than in S2_NOCHEM (Fig. 2). Furthermore, a larger cooling over the Barents Sea is found in S2_CHEM, which extends towards Northern Eurasia. In the S1 experiments temperature and sea ice anomaly patterns are weaker but similar to S2 and S1_CHEM is characterized by an amplified cooling as well (not shown). During the first 10 years, when no AMOC differences between the CHEM and NOCHEM experiments are found, the temperature and sea ice anomalies are very similar. The Arctic sea ice differences between CHEM and NOCHEM, which emerge in the last 20 years

of the reduction period, are therefore related to the weaker AMOC in the CHEM experiments and the reduced heat transport into the Arctic.

The cooling in the lower atmosphere has a direct effect on the ocean. With a reduction of the upper ocean temperatures and an increased salinity due to the enhanced sea ice formation, the density of the upper ocean increases almost everywhere (Fig. 3a-i).
Additionally, a shift of the storm track and a significant reduction of the precipitation in the North Atlantic contributes to the salinity and density increase (not shown). During the first 10 years of the SRR period, differences in the density anomalies in the upper ocean of the North Atlantic are small and not significant, except for a region South of Greenland, where the density is significantly higher in S2_NOCHEM (Fig. 3a-c). In the following decade further increases of the upper ocean density are found in both experiments, but the anomalies are again larger in S2_NOCHEM (Fig. 3d-f). Now, the density anomalies in large parts of the North Atlantic are more pronounced in S2_NOCHEM in comparison to S2_CHEM. In the last 10 years, finally, density anomalies are still strongly positive, but the differences between both experiments weaken (Fig. 3g-i).

Convection takes place in the Nordic Seas and in a region in the North Atlantic close to the Labrador Sea (contours in Fig. 3e-h). The intensity of the deep water formation in these two regions is an important driver of AMOC variability (Jungclaus et al., 2005). Focusing on the changes in the Nordic Seas, we find an intensification of the deep water formation already for the first 10 years of the reduction period (Fig. 3j-l). A further intensification is found for the second and the third decade, but the anomalies between S2_CHEM and S2_NOCHEM show only weak significance. The anomalies in the S1 experiments are similar, i.e., differences are mostly non-significant. Density changes in the Nordic Seas are driven by a combination of temperature and salinity changes (Fig. 4). The temperature changes, however, dominate in the first half of the SRR period, while the increasing salinity drives the density changes in the second half.

In the North Atlantic the density and mixed layer differences between S2_CHEM and S2_NOCHEM are larger than the ones in the Nordic Seas. During the first 10 years of the SRR period, positive mixed layer depth anomalies are found in S2_NOCHEM (Fig. 3k), while no consistent response is found in S2_CHEM (Fig. 3j). Consequently, the intensification is significantly stronger in S2_NOCHEM (Fig. 3l). A similar picture emerges for the second decade (Fig. 3m-o). In the third decade a clear intensification is obvious in S2_CHEM, while a slight reduction is found in S2_NOCHEM in the southern region of the North Atlantic convection zone (Fig. 3p-r). As in the Nordic Seas, the density changes are driven by the reduced temperatures in the first half of the SRR (Fig. 4). In the second half of the SRR period the salt content of the upper ocean increases, while temperatures increase again, related to the intensification of the overturning. The salinity changes, nevertheless, leads to a further increase of the density in the second half of the reduction period.

The increasing density and deep water formation in both convective regions help to understand the intensification of the AMOC in the course of the SRR. Driven directly by the temperature response to the reduced solar forcing, this mechanism can be considered as the thermal effect of the SRR on the overturning. However, in S2_CHEM the intensification of the convection in the North Atlantic is delayed in comparison to S2_NOCHEM. Similar differences are found between the two S1 experiments (Fig. 1c). A further mechanism is therefore needed to understand the differences in the AMOC response between the CHEM and NOCHEM experiments.

## 3.2 The dynamical effect and the role of chemistry-climate interactions

Chemistry-climate interactions are most pronounced in the stratosphere (e.g., Dietmüller et al., 2014). In particular, the different response of the stratospheric ozone and water vapour between CHEM and NOCHEM (Fig. S1) leads to large differences in the stratospheric temperatures. For S2_CHEM temperature anomalies of up to –28 K are found in the upper stratosphere (Fig. 5a). Above 1 hPa the maximum cooling is found in the polar latitudes with a second maximum in the tropics. In the lower and middle stratosphere, the cooling is stronger in the tropics and mid-latitudes. With about –10 K, the maximum cooling in S2_NOCHEM is much smaller than the response in S2_CHEM (Fig. 5b,c). Furthermore, as a consequence of the missing response of the ozone concentrations to the reduced solar forcing, the effect of the lower and middle stratospheric cooling on the meridional temperature gradient is weaker.

The response of the zonal mean wind in the stratosphere agrees well with the temperature anomalies. For S2_CHEM, a pronounced weakening of the NH and SH polar vortices is found (Fig. 5d-f). Using the zonal mean wind component at 60°N and 10 hPa as index for the intensity of the NH polar vortex (Christiansen, 2001, 2005), a reduction of –43 % is found in S2_CHEM during the winter season (Nov. to Mar.) when averaged over the SRR period. The largest wind anomalies occur during the vortex maximum in January. The reduction in S2_NOCHEM is much weaker (–8 %) than in S2_CHEM. Furthermore, the duration of the winter period with predominant westerly wind is reduced in S2_CHEM by –30 % and in S2_NOCHEM by –5 % respectively, when defining the start of the winter period by the day with the first occurrence of a westerly daily mean zonal mean wind component at 60N and 10hPa after September and the end by the first day with easterly winds after March. Qualitatively similar results are found for the S1 experiments, with NDJFM vortex anomalies of –9 % for S1_CHEM and –2 % for S1_NOCHEM (Fig. S3). These responses highlight the non-linear relationship between the solar forcing and the atmospheric dynamics.

The weakening of the NH polar vortex is closely related to the occurrence of sudden stratospheric warming (SSW) events (Fig. 6). SSWs are stratospheric extreme events, in which the westerly flow during winter time is reversed and a strong warming in the polar stratosphere is observed. SSW events in the NH are associated with a break down of the polar vortex. Following the SSW definition by Charlton and Polvani (2007) almost a doubling of the number of SSW events is found in S2_CHEM (1.34 events/winter in comparison to 0.68 events/winter in CTRL_CHEM). In S1_CHEM an increase to 0.73 events is simulated. Similarly to the NH polar vortex, the effect of the SRR on the SSW events is small in NOCHEM. For S1 the average number of events increases from 0.68 events/winter in CTRL_NOCHEM to 0.70 events/winter in S1_NOCHEM. In S2_NOCHEM an increase to 0.73 events is simulated. While the increase in the mean number of SSW events is small in S2_NOCHEM, a clear reduction of the years with a low number of SSW events is found (lower quartile of the boxplot).

The NH polar vortex and extreme events like SSW affect the tropospheric circulation in the NH by stratosphere-troposphere interactions. A downward propagation of wind speed anomalies from the middle stratosphere to the surface is related to positive and negative phases of the AO (Baldwin and Thompson, 2009). For a negative phase of the AO, negative wind anomalies in the stratosphere occur up to 40 days before the AO event takes place at the surface (Fig. S4). For a positive phase of the AO,

the zonal wind anomalies are even stronger (not shown). Overall, the downward propagation of wind speed anomalies does not differ substantially between the CHEM and NOCHEM control experiments.

The stratospheric changes in the course of the SRR therefore affect the tropospheric pressure systems. In Figure 2a the sea level pressure anomalies for S2_CHEM reveal a pattern of positive anomalies over large parts of the Arctic and negative anomalies in the North Atlantic and the Northern Pacific, similar to a negative phase of the AO. In S2_NOCHEM comparable negative and positive pressure patterns are found, but the anomalies are much weaker (Fig. 2b). Due to the strength of the response, the winter phenomenon AO is reflected in the annual mean values (Fig. 2). However, when focusing on the winter season (Nov. to Mar.) and the AO index, the strength of the anomalies in S2_CHEM is even more apparent (Fig. 7). During the entire SRR phase a persistent negative phase of the AO is found in S2_CHEM. In S1_CHEM the tendency towards a negative AO is found as well, although the response is weaker and several years with a positive phase of the AO occur during the SRR. In the NOCHEM experiments the response is in general weaker, but a shift towards a negative AO phase from CTRL_NOCHEM to S1_NOCHEM and S2_NOCHEM is apparent (Fig. 7d). In particular, negative AO phases tend to occur more often in the first half of the SRR period, while neutral conditions dominate in the second half.

Atmospheric chemistry-climate interactions therefore lead to pronounced differences in the dynamical response to the SRR, from the stratosphere down to the surface of the NH high latitudes. With a shift in the pressure pattern which affect the wind systems in the lower atmosphere, these differences have the potential to also modify the oceanic circulation.

The control experiments are used to assess the influence of the AO phase on the North Atlantic. Regressing the AO index on different oceanic variables reveals that a negative AO phase is associated with an increased downward heat flux south of Greenland and negative heat flux anomalies close to the east coast of North America during winter in CTRL_CHEM (Fig. 8a). Sea ice cover is reduced in the Labrador Sea (Fig. 8b) and the dynamical changes lead to an increased total freshwater flux into large parts of the North Atlantic, and a reduced flux in the Nordic Seas (Fig. 8c). These changes cause a reduction of the salinity (Fig. 8d), except for a small region South of Greenland, which may be affected by a weakening of the East Greenland current. Additionally, SSTs increase South of Greenland (Fig. 8e), related to the enhanced downward heat flux. Since the density of the water decreases with increasing temperature and decreasing salinity, all these changes lead to a pronounced reduction of the mixed layer depth (Fig. 8f). In CTRL_NOCHEM the effect of the AO is very similar (Fig. S5).

These changes at the ocean surface are also reflected in the AMOC index. In both control experiments the AMOC reacts within the same winter season to the AO phase, as detected by the positive correlation between the winter AO and the AMOC index of the same season (Fig. S6). Furthermore, the AO phase has long lasting effect on the overturning, reflected in significant positive correlations for lags up to 9 years.

To summarize, the weaker intensification of the AMOC in the CHEM experiments, in comparison to NOCHEM, is related to a second (dynamical) response to the SRR. With interactive chemistry, the stratospheric cooling is strongly amplified by stratospheric ozone loss. As a consequence, the weakening of the Northern polar vortex is more pronounced, which has larger effects on the tropospheric circulation patterns, in particular the phase of the AO. The dynamical changes decrease the density of the surface ocean waters South of Greenland, reduce convection, and weaken the AMOC. In the NOCHEM experiments a

tendency towards a negative phase of the AO is found as well, but less pronounced, due to the absence of chemistry-climate interactions. The dynamical effect on the AMOC is therefore much weaker and the thermal response dominates.

## 4 Conclusions and Discussions

Sensitivity experiments for different solar minima and model configurations with and without chemistry-climate interactions have been carried out to study the response of the AMOC to reduced solar forcing and the modulating role of chemistry-climate interactions. Without interactive chemistry the response of the AMOC is dominated by the direct thermal effect leading to an intensification of the overturning circulation. A second dynamical effect is identified in the experiments with chemistry-climate interactions and leads to an weakening of the overturning.

The two processes are summarized in Figure 9: The thermal effect is related to the reduced short-wave energy reaching the troposphere and the surface and the ensuing cooling of the lower atmosphere and the upper ocean. This increases the sea surface density and enhances convection. The thermal effect, however, is compensated by the dynamical effect when atmospheric chemistry is taken into account. Induced by the reduction of the tropical stratospheric temperatures, a weakening of the NH polar vortex and – by interactions between the stratospheric and tropospheric circulation – a negative phase of the AO, is found in response to the SRR. The circulation changes in the troposphere in turn cause a weakening of the AMOC by anomalous heat and freshwater fluxes. The dynamical effect is amplified by chemistry climate interactions, due to the enhanced stratospheric temperature response related to the effect of the reduced UV radiation on the ozone concentrations. For the weaker S1 SRR, both effects cancel each other and therefore no AMOC intensification is found in the experiments with interactive chemistry. In the S2 experiments with stronger forcing, however, the thermal response of the AMOC dominates and the dynamical effect leads only to a reduced intensification of the overturning.

The thermal effect of solar radiation changes on the overturning has been identified in earlier studies (Cubasch et al., 1997; Latif et al., 2009; Otterå et al., 2010; Swingedouw et al., 2011). Related to increasing global greenhouse gas concentrations and associated surface warming, it is also one of the dominant mechanisms for the projected future weakening of the AMOC (Stocker and Schmittner, 1997; Manabe and Stouffer, 1999; Mikolajewicz and Voss, 2000; Gregory et al., 2005; Stocker et al., 2013). The thermal response to the reduced solar forcing has also implications for the projected weakening of the AMOC in the 21th century. Several studies suggest that the Sun may enter a grand solar minimum within the next 100 years (Lockwood et al., 2009; Steinhilber and Beer, 2013; Roth and Joos, 2013), although the amplitude of the TSI changes is associated with large uncertainties. While the influence on the global mean temperature increase is small (Feulner and Rahmstorf, 2010; Meehl et al., 2013; Anet et al., 2013b), the thermal effect may reduce the projected 21th century AMOC weakening. This is confirmed by experiments of Anet et al. (2013b). The AMOC is significantly stronger in the late 21th century, in ensemble simulations simulations including a grand solar minimum in the second half of the 21th century in comparison to ensemble simulations without a decline of the solar activity (Fig. S7).

Parts of the dynamical effect have been reported in previous studies. The relationship between solar variability and the stratospheric circulation was found for the 11-yr cycle (Kodera and Kuroda, 2002; Mitchell et al., 2015) as well as for grand

solar minima (Anet et al., 2013a). Also the projection of the stratospheric anomalies on the AO was reported in previous studies (Kodera, 2003; Ineson et al., 2011; Scaife et al., 2013). Finally, the influence of the AO phase on the overturning was studied (Delworth and Greatbatch, 2000; Eden and Willebrand, 2001; Matthes et al., 2006; Delworth and Zeng, 2016) and a few studies identified a possible influence of the stratospheric circulation on the overturning (Manzini et al., 2012; Reichler et al., 2012).

The stratosphere responds very rapidly to the reduced solar forcing and the tropospheric AO index shifts to a negative phase in the second winter after the onset of the reduction period, although it takes about 5 years, before a persistent negative AO phase is found in S2_CHEM. The response of the AMOC, however, is delayed by several years. A similar delay was reported by Delworth and Zeng (2016) who performed sensitivity experiments with an ocean model forced by different atmospheric conditions. In one experiment a persistent positive phase of the NAO is simulated and the AMOC responds to this forcing with

strengthening of the circulation, which is delayed by 5-7 years (compare Fig. 3 in Delworth and Zeng, 2016). This lag of the response agrees with our results, although an exact timing is difficult to estimate from our setup.

The influence of the dynamic effect on the AMOC may furthermore depend on the length of the solar reduction period. Lohmann et al. (2009) found a gradual weakening of the subpolar gyre response with time in ocean model simulations forced with a persistent negative phase of the NAO. Additionally, the response of the AMOC may be non-linear and an increase of the

solar forcing may change the dynamic effect (Lohmann et al., 2009).

Recently, Chiodo and Polvani (2016) assessed the role of the interactive chemistry on the temperature and precipitation response to increasing SSI. They identified a reduced sensitivity with interactive chemistry due to the effect of the ozone increase on the short-wave radiation balance. Our results for a SSI reduction indicate a slightly larger temperature sensitivity with interactive chemistry owing to the effect of the stratospheric water vapour and ozone changes on the long-wave radiation

balance. These differences may be attributed to model differences or differences in the response of the climate system to increasing and decreasing solar forcing. A possible effect of the differences in the atmospheric response on the AMOC is not discussed by Chiodo and Polvani (2016).

Here, we show for the first time how stratospheric processes modulate the modelled response of the AMOC to solar forcing and identify the importance of chemistry-climate interactions for the response. Hence, previous studies without atmospheric

chemistry may overestimate the sensitivity of the AMOC to solar forcing, since the dynamical effect is absent.

Furthermore, our results reveal possible additional side effects of the solar radiation management technique: A reduction of the incoming solar radiation in space to mitigate the temperature increase caused by the emission of GHGs might affect the tropospheric circulation patterns in the NH and cause a weakening of AMOC with climatic consequences, in particular for the temperate climate in western Europe. The dynamical effect is expected to change, however, when the solar radiation is reduced

in the Earth's atmosphere, for instance, by stratospheric sulphate aerosols. In this case, a strengthening of the NH polar vortex and a positive phase of the AO may develop, analogous to the response to strong tropical volcanic eruptions (Graf et al., 1993; Kodera, 1994; Stenchikov et al., 2002; Muthers et al., 2014a, 2015). This effect of the positive AO phase may, in turn, lead to an intensification of the AMOC. Future studies shall address the influence of stratospheric sulphate geoengineering on the AMOC and the possible role of chemistry-climate interactions.

*Acknowledgements.* We thank the four anonymous reviewers for their constructive comments. This work has been supported by the Swiss National Science Foundation under grants CRSII2-147659 (FUPSOL II) and 200020-159563.

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

**Table 1.** Overview of the ensemble experiments used in this study. Each ensemble consists of 10 experiments.

| Experiment | TSI [$\mathrm{Wm^{-2}}$] | Chemistry |
|---|---|---|
| CTRL_CHEM | const. | Yes |
| CTRL_NOCHEM | const. | No |
| S1_CHEM | –3.5 | Yes |
| S1_NOCHEM | –3.5 | No |
| S2_CHEM | –20 | Yes |
| S2_NOCHEM | –20 | No |

Swingedouw, D., Terray, L., Cassou, C., Voldoire, A., Salas-Mélia, D., and Servonnat, J.: Natural forcing of climate during the last millennium: Fingerprint of solar variability, Climate Dyn., 36, 1349–1364, doi:10.1007/s00382-010-0803-5, 2011.

Swingedouw, D., Ortega, P., Mignot, J., Guilyardi, E., Masson-Delmotte, V., Butler, P. G., Khodri, M., and Séférian, R.: Bidecadal North Atlantic ocean circulation variability controlled by timing of volcanic eruptions, Nature Communications, 6, 6545, doi:10.1038/ncomms7545, 2015.

Thompson, D. W. and Wallace, J. M.: Regional climate impacts of the Northern Hemisphere annular mode., Science, 293, 85–9, doi:10.1126/science.1058958, 2001.

Valcke, S.: The OASIS3 coupler: A European climate modelling community software, Geosci. Model Dev., 6, 373–388, doi:10.5194/gmd-6-373-2013, 2013.

Waple, A. M., Mann, M. E., and Bradley, R. S.: Long-term patterns of solar irradiance forcing in model experiments and proxy based surface temperature reconstructions, Climate Dyn., 18, 563–578, doi:10.1007/s00382-001-0199-3, 2002.

Wunsch, C.: Oceanography. What is the thermohaline circulation?, Science, 298, 1179–81, doi:10.1126/science.1079329, 2002.

Zanchettin, D., Timmreck, C., Graf, H.-F., Rubino, A., Lorenz, S., Lohmann, K., Krüger, K., and Jungclaus, J. H.: Bi-decadal variability excited in the coupled ocean–atmosphere system by strong tropical volcanic eruptions, Clim. Dyn., 39, 419–444, doi:10.1007/s00382-011-1167-1, 2012.

Zubov, V., Rozanov, E., Egorova, T., Karol, I., and Schmutz, W.: Role of external factors in the evolution of the ozone layer and stratospheric circulation in 21st century, Atmospheric Chemistry and Physics, 13, 4697–4706, doi:10.5194/acp-13-4697-2013, 2013.

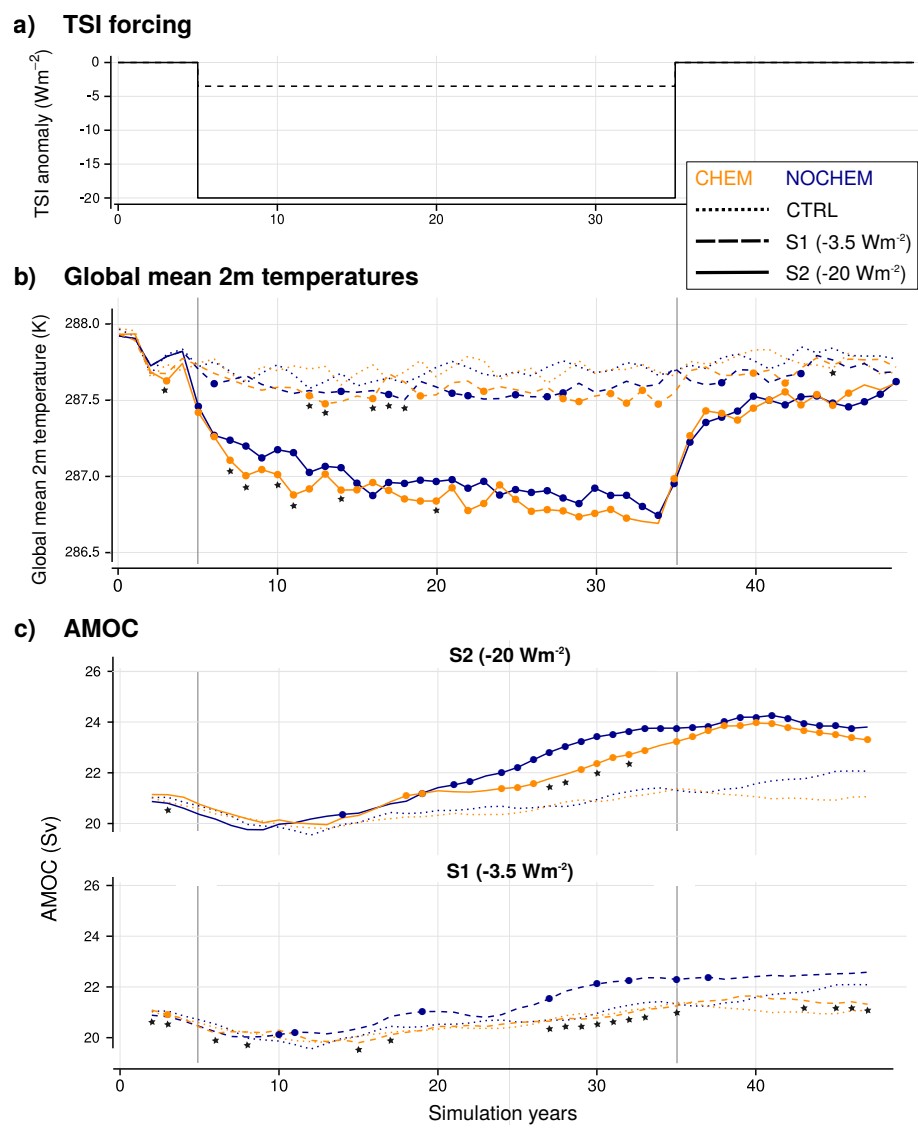

**Figure 1. a:** Total solar irradiance (TSI) anomaly of –3.5 $\mathrm{Wm}^{-2}$ (dashed) and –20 $\mathrm{Wm}^{-2}$ (solid) applied in this study. **b:** Global annual mean ensemble mean 2-m temperature. **c:** Ensemble mean AMOC index in the different experiments, smoothed using a 5-yr running mean. Dots denote significant differences in the (un-smoothed) annual mean values between the SRR ensemble and the control ensemble (Student's t-test, $p \leq 0.05$). Small stars below the CHEM time series correspond to years with significant differences between the CHEM and NOCHEM experiment ($p \leq 0.05$). The beginning and the end of the SRR period is indicated by vertical lines in panels b and c.

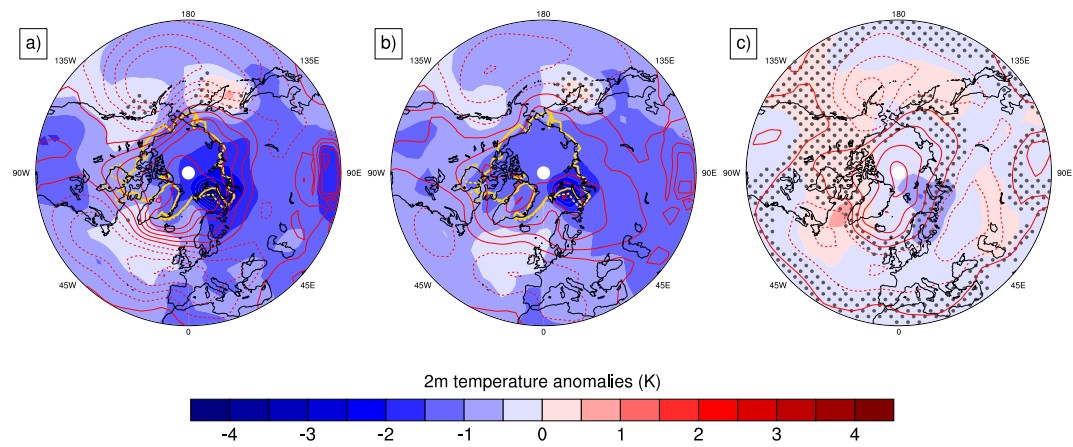

**Figure 2.** Annual mean 2-m temperature anomalies (colors), sea level pressure anomalies (red contours), and 50 % sea ice extent line (yellow contours) averaged over the SRR period. Temperature and sea level pressure anomalies are calculated relative to the control ensemble mean, for the sea ice extent the values of the control ensemble and the S2 experiments are depicted by the solid and dashed line, respectively. **a:** shows the difference for the S2_CHEM ensemble, the S2_NOCHEM anomalies are shown in **b**. Panel **c** displays the differences between S2_CHEM and S2_NOCHEM. Dots denotes non-significant temperature differences (Students t-test, p>0.05). The sea level pressure contour interval is 0.25 hPa and negative sea level pressure anomalies are dashed.

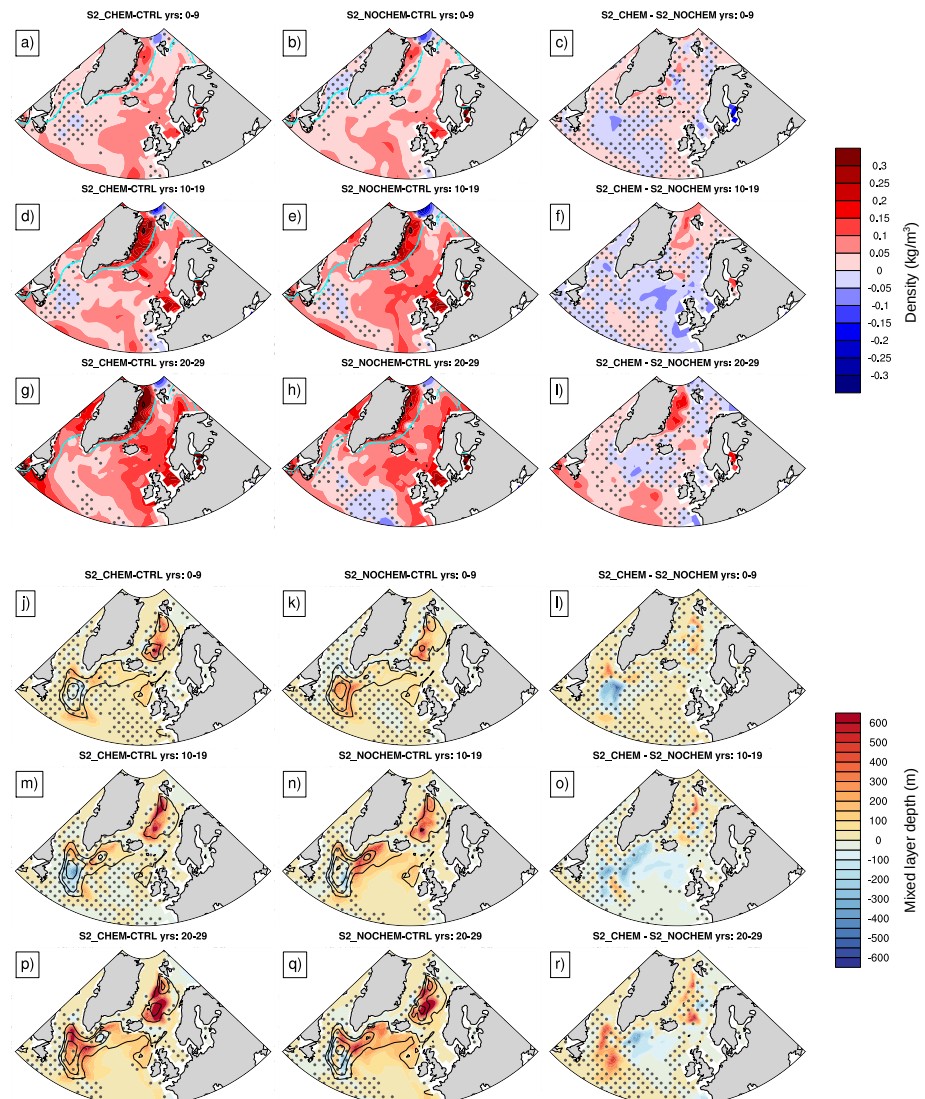

**Figure 3. a-i**: S2_CHEM (a,d,g), S2_NOCHEM (b,e,h), and the difference between S2_CHEM and S2_NOCHEM (c,f,i) ensemble mean upper ocean (0–220 m) density anomalies (kg/m$^3$) for late winter (Jan.-Mar.) averaged over the first (a-c), second (d-f), and last decade (g-i) of the SRR period. Cyan contours display the extend of the 50 % sea ice area for the CTRL ensemble mean (solid line) and the SRR experiments (dashed line). **j-r**: S2_CHEM (j,m,p), S2_NOCHEM (k,n,q), and the difference between S2_CHEM and S2_NOCHEM (l,i,r) ensemble mean Jan.-Mar. mixed layer depth anomalies (m, shading) averaged over the first (j-l), second (m-o), and last decade (p-r) of the SRR period. Contours denoted the average Jan.-Mar. average mixed layer depth in CTRL_CHEM and CTRL_NOCHEM, respectively, with a contour step of 500 m. Dots denotes non-significant density or mixed layer depth differences (Students t-test, p>0.05).

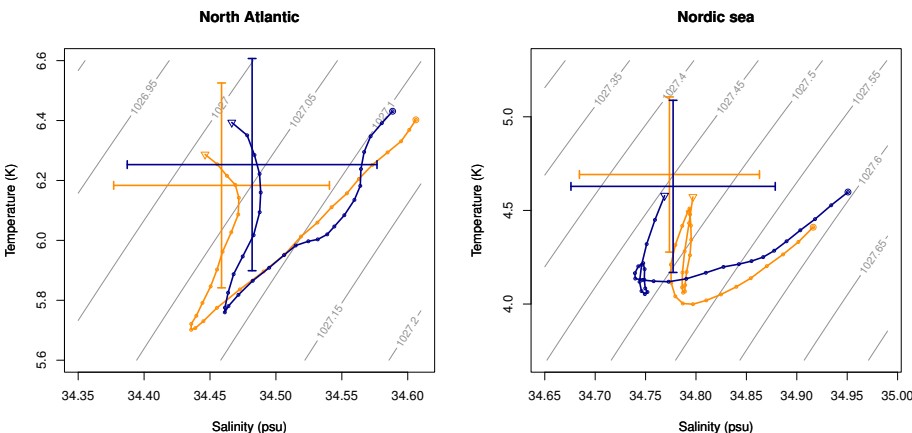

**Figure 4.** Temperature salinity averaged over the upper 220 m for the two deep water formation region North Atlantic and Nordic Seas. The deep water formation regions cover all grid cells with an annual mean mixed layer depth $\geq 250$ m in the corresponding ocean basins. The lines show the salinity and temperature development from the beginning (triangle) to the end (large dot) of the SRR for the S2_CHEM (orange) and S2_NOCHEM (blue) experiments. Each point represent a single year. To improve visibility, the values are smoothed using a 15-yr low pass filter. Error bars denote the mean and the standard deviation of the corresponding control ensembles. Contours represent the water density.

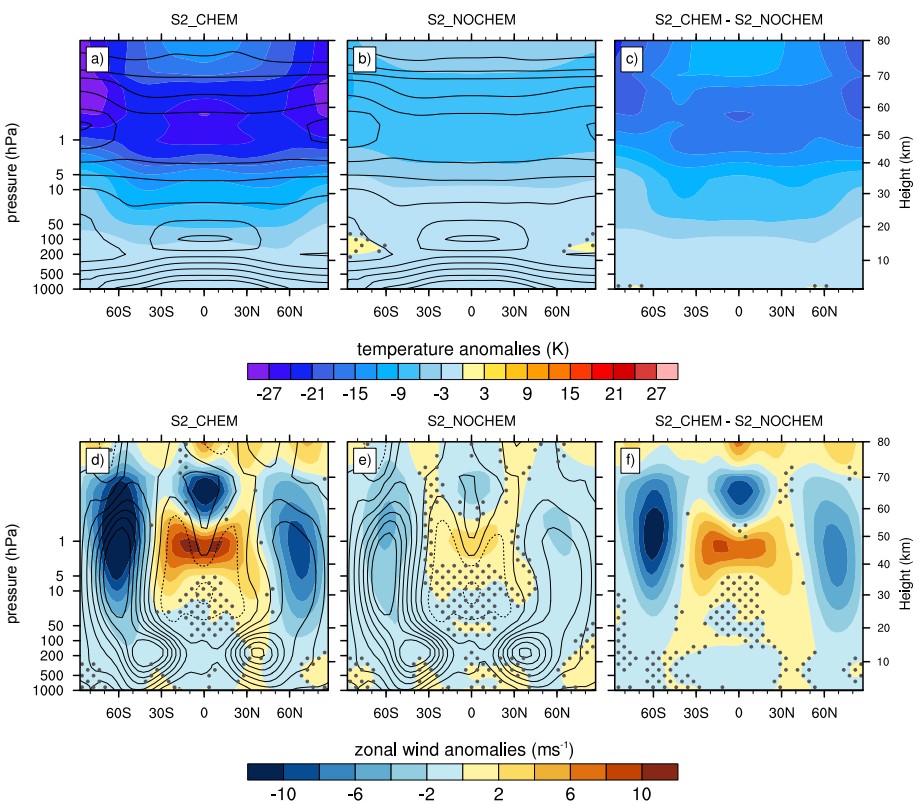

**Figure 5.** Annual mean zonal mean temperature (**a-c**) and annual mean zonal mean zonal wind (**d-f**) anomalies in the S2 experiments relative to the control experiments: **a,d** shows the anomalies for the S2_CHEM experiment and **b,e** the results for S2_NOCHEM. The differences between both experiments (S2_CHEM - S2_NOCHEM) are shown in **c,f**. Anomalies are averaged over the 30-yr SRR period. Contours represent the mean state in the control experiments with contours from 180 to 280 K (contour step 15 K) for the temperatures and -30 to 30 ms$^{-1}$ (contour step 5 ms$^{-1}$) for the zonal wind. Dots denotes non-significant temperature differences (Students t-test, p>0.05).

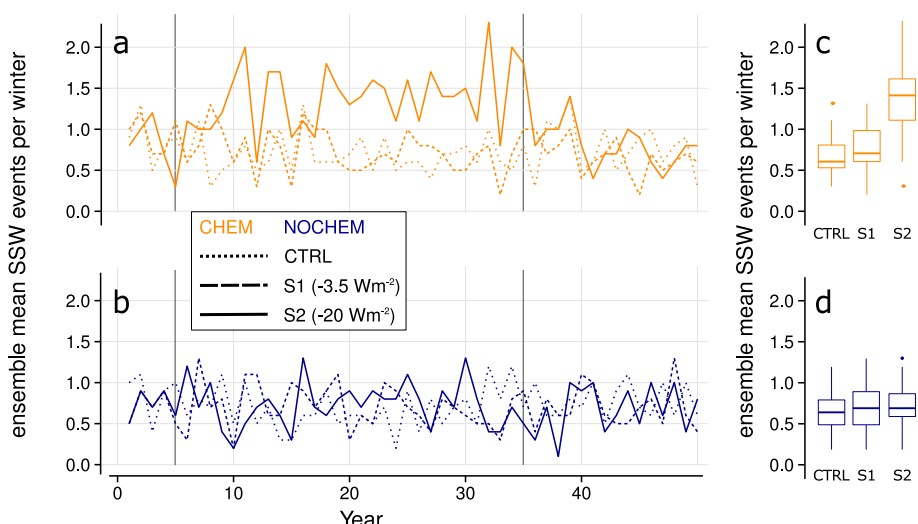

**Figure 6. a-b:** Ensemble mean number of sudden stratospheric warming events (SSW) per winter season (Nov. to Mar.) as in defined by Charlton and Polvani (2007). **c-d:** Boxplot statistics for the number of SSW events per winter season averaged over the SRR period. The beginning and the end of the SRR period is indicated by vertical lines in panels a and b.

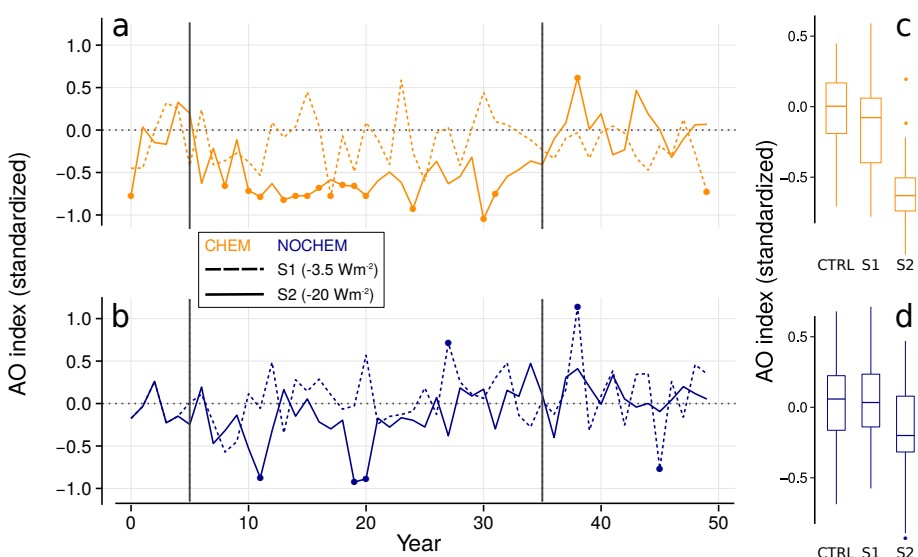

**Figure 7. a-b:** Ensemble mean AO index (standardized and reversed sea level pressure difference between $45°$N and $65°$N) per winter (Nov. to Mar.). Dots indicate winters with significant differences to the CTRL ensemble (Student t-test $p \leq 0.05$). **c-d:** Boxplot statistics for the AO index averaged over the SRR. The beginning and the end of the SRR period is indicated by vertical lines in panels a and b.

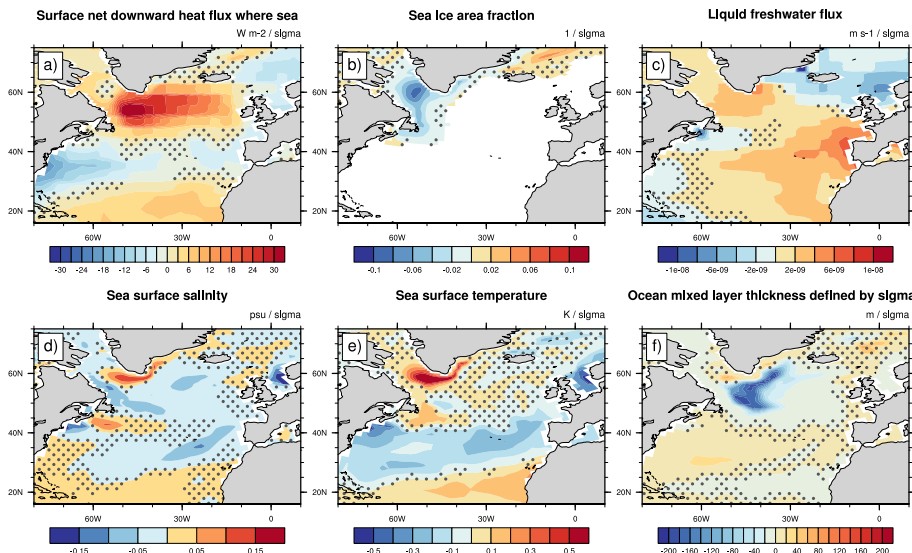

**Figure 8.** Influence of a negative AO phase on different oceanic variables in CTRL_CHEM during winter (Nov. – Mar.). Linear regression coefficients for **(a)** net downward heat flux, **(b)** sea ice area fraction, **(c)** liquid freshwater flux (evaporation minus precipitation), **(d)** sea surface salinity, **(e)** sea surface temperature, and **(f)** mixed layer depth. To highlight the influence of a negative AO phase the AO index has been reversed in the regression analysis. Dots denotes non-significant temperature differences (Students t-test, $p > 0.05$).

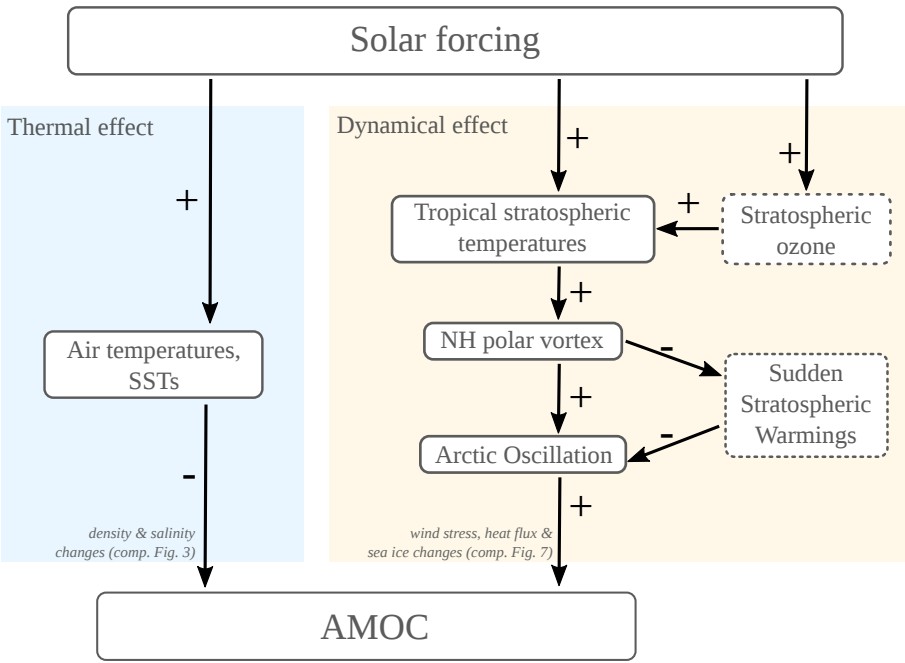

**Figure 9.** Flowchart summarizing the thermal and dynamical effect of a change in solar radiation on the AMOC. The sign indicate the correlation between two processes. Dashed boxes represent effects, which are amplified by chemistry-climate interactions.