# Peer review of "Response of the AMOC to reduced solar radiation – the modulating role of atmospheric-chemistry"

_Earth System Dynamics, 2016_

## Referee Comment (RC1) · Anonymous Referee #1 · 22 May 2016

This is a mostly well written manuscript with interesting new results identifying a stratospheric mechanism impacting the Atlantic Meridional Oscillation. Therefore, it is potentially suitable for publication in the Earth System Dynamics journal. I have, however, a few concerns which I would like the authors to address before I can recommend the publication.

My major concerns are:

1. What would be the impact of aerosols? Your model does not include aerosol interactions, you just simply reduce the solar radiation. This seems a critical simplification to me. You should at least discuss how aerosol interactions would modify the AMOC response if taken into account in your model.

2. I think in reality the salt rejection from the sea-ice growth is rather small and mainly

occurs north from the regions of deep convection. Therefore it has only a minor importance to the deep convection and the AMOC compared to the heat loss and possibly the net precipitation (precipitation minus evaporation) at the ocean surface. At the moment, the reader is led to understand that the salt rejection is at least as important as the heat loss. The increase of the sea-surface salinity could also be due to a decreased net precipitation related to changing storm tracks, for example. To better support the salt rejection argument, you need to quantify the salt rejection to the surface density and compare it to other factors. Can you check the ocean surface fluxes from your model output and their relation to the T and S, not only density, anomalies? How realistic these modelled fluxes then are, depend on your model skill and are related to your model configuration, such as the sea-ice salinity scheme.

3. I have a problem when you treat the AO and NAO identically. Although the AO and NAO correlate, they are not identical, not even from the AMOC perspective. I agree that the AO behaves largely like the NAO in winter. If, instead of the AO, you based your analysis on the NAO, how would the results look like? What would be their significance after taking into account the possible year-to-year autocorrelation?

Minor comments:

- Page 1, line 16. I would rather say that 'surface currents transport water into the northern North Atlantic' rather than to 'Northern high latitudes' which sounds more like to the Arctic Ocean.

- Page 2, line 7. I don't think the AO is the hemispheric equivalent of the NAO. The NAO is a regional index and correlates with the AO, but their definitions differ substantially.

- Page 2, lines 18. '... by increasing SSTs and enhancing freshwater input ...'

- Page 2, line 32. As you focus on the AO in this paper, would be clearer not to talk about the NAO, but the AO, after Page 2, line 7.

- Page 3, line 19. '... uses temperature data ...'

- Page 4. line 3. You provide very little details on the model configuration. For example, what was the time step you used? How about the sea-ice salinity, was it constant? Or what sea-ice thermodynamics scheme was deployed? This information is important to assess how realistically the sea-ice salt rejection was modelled.

- Page 4, line 20. You should mention here how long model simulations continued after the 30 year SSR period.

- Page 4, line 27. Explain the acronym TSI.

- Page 4, line 32. Explain more in detail how the AO index was calculated and provide references. For example, a common way to calculate the AO is based on the PC1 of 1000 mb pressure height anomaly data north of 20N. Your method seem to differ from that. Why? How robust your results are based on the AO calculation method?

- Page 5, line 18. '... are related ...'

- Page 5, lines 21-22. This sentence is hard to understand. How is the slight initial reduction of the global mean temperature related to the initial conditions of the ocean when the ocean initial conditions are from a 1300 year long simulation? Why rather not related to the atmospheric initial conditions which presumably started from an observation based, physically less consistent initial state?

- Page 5, line 31. 'during the second half'

- Page 6, line 7. Do you mean that sea-ice patterns look similar but their anomalies are (presumably) weaker in the S1 experiments?

- Page 6, line 13. Is the reduction in precipitation related to a shift in the main storm track and, as a result, a colder and dryer atmosphere?

- Page 6, lines 15-16. I think you need to verify the significance of the salt rejection to the surface density. It is typically small compared to the cooling effect. Also, not much freezing occurs at the eastern side of Greenland, but the Arctic ice flows south and

melts along the eastern boundary of the East Greenland Current.

- Page 6, lines 26-27. You should mention that these density and mixed layer depth anomalies are not reflected in the AMOC.

- Page 6, line 30. '... the North Atlantic (Fig. 3a).'

- Page 6, lines 33-34. You must mean 'the central North Atlantic' here.

- Page 7, line 2. The 'dominance' is based on very speculative assumptions. Just say 'Salinity changes, nevertheless ...'

- Page 8, line 1. Add a literature reference that proofs the linkage between the downward propagating wind anomalies and the AO phases.

- Page 8, lines 13-14. You don't show this in Fig. 6, which should be mentioned, or plot CTRL_NOCHEM in Fig. 6.

- Page 8, line 16. '... which affects the wind ...'

- Page 8, line 29. '... the AO phase has a long lasting effect ...'

- Page 8, line 33. This should be '... the weakening of the Northern polar vortex ..', right?

- Page 8, lines 34-35. I suggest you to write '... dynamical changes decrease the density of the surface ocean waters South of Greenland, ...'

- Page 9, lines 11-13. Don't these citations analyse the impact of the increase in GHGCs? Seems like you are cutting corners here. Wouldn't it be more correct to say e.g. '... Swingedouw et al., 2011). Related to increasing global greenhouse gas concentrations and associated surface warming, it is also one of the dominant ...'

- Page 9, line 17. '... may reduce the projected 21st century ...'

- Page 9, line 18. '... stronger than in the late 21st century than [today?], when a grand ...'

- Page 9, line 23. '... the AMOC by anomalous ...'

- Page 9, lines 23-25. This sentence is not clear to me. I suggest rewriting 'The dynamical effect is enabled by chemistry climate interactions, which result in amplified stratospheric temperature responses.'

- Page 9, lines 28-29. The literature you cite here include three studies analysing NAO and only one analysing AO. This indicates to me, that NAO would have been a more appropriate index for this study as well, although its relation to the polar vortex is not as clear as the one of the AO.

- Page 9, line 31. 'the modelled response of the AMOC ...'

- Page 10, line 5. '... weakening of the AMOC with climatic ...'

- Page 10, line 10. What do you mean by 'Future studies'. Be more explicit. Are you planning to do this work?

- Figure 1. Write out the TSI acronym in the figure caption. As you used t-test for significance, did you check the autocorrelation or did you just treat each year as an independent variable? If years correlate, it affects your significance estimates. Explain more in detail what you did.

- Figures 2-4, 7, S1, S3, S5. Dots are not dark grey, but black. Better to say 'Black dots denote non-significant ...'

- Figure 2. More correct to say 'The sea-level pressure contour interval is ...'

- Figure S2. Indicate latitude and longitude locations of these T & S profiles.

---

## Referee Comment (RC2) · Anonymous Referee #2 · 26 May 2016

Review of „Response of the AMOC to reduced solar radiation – the modulating role of atmospheric-chemistry" by S. Muthers et al.

This paper examines the physical processes responsible for the AMOC response to reduced solar radiation and assesses the importance of chemistry-climate in modulating this response. By comparing two sets of climate model experiments, with and without interactive chemistry, the study demonstrates that climate models which do not consider stratospheric - namely ozone - chemistry may overestimate the sensitivity of the AMOC to solar forcing since the "top-down" influence (stratospheric influence on tropospheric circulation) is underestimated.

In my opinion, this work constitutes a very nice contribution for the broad climate research readership as it demonstrates, using the specific example of the AMOC, the prominent and complex connections between the different components which drive climate variability (going from the stratosphere chemistry to the ocean circulation). The work is well-framed in the current literature. I found the paper mostly clear, well written and scientifically sound. I think however that some improvements and clarifications could be made before publication. Please find my main comments/suggestions below:

**Main comments/questions**:

1/- It has already long been recognized that atmospheric chemistry interacts with dynamics and that its consideration in climate models is crucial to adequately simulate climate variability (e.g. influence of the ozone hole recovery on the SAM trends in CMIP3 simulations by Son et al. (2008)). As a consequence, historical and projection climate simulations in CMIP5 for models without interactive chemistry were designed by prescribing chemical fields that consider long-term trends (Cionni et al., 2011). For CMIP6, the ozone prescription fields should be even further improved. So I would say that the current question regarding chemistry-climate interactions is: do we really need interactive chemistry? or can it just be prescribed? The other question is then how to prescribe it in the most accurate way (see e.g. Nowack et al. (2015)).

In my opinion, given the frame, the results and the conclusion of the present study, I think that the introductory part of the paper should – at least partly – review the recent advances regarding chemistry climate interaction. A lot has been done already and should not be ignored.

2/- In the light of my previous comment, I would suggest the authors to explain more thoroughly how the combined UV+ozone effects modulate the heating rates in the stratosphere which is the starting point of the stratospheric mechanism discussed in the paper. The thermal modulation of the stratosphere through UV variations comes from two main effects: (1) direct shortwave heating through incoming UV absorption by ozone ($\lambda \sim 200\text{-}300$ nm), (2) ozone change ($\lambda < 242$ nm) which also affect shortwave heating rates. Both effects count significantly. Basically, and if I understood correctly, their NOCHEM experiment account for effect (1) only while CHEM account for effects (1) + (2). I think such clarifications are easy to make and necessary since they help understanding the basic difference between the two experimental configurations (at least

regarding stratospheric ozone which is the major solar effect). In the present version of the paper too few information are given on UV-ozone-temperature interactions and their implication on experimental setting (e.g. P2L32-P3L1, P5L12-14).

3/- A very recent study by Chiodo and Polvani (2016) has just been released in *Journal of Climate* and deal with – somewhat - similar problematics. They performed simulations that also present some similarities with those perform in the present work. While both studies have their own relevance and focus on different aspects, they also nicely complement each other. The authors may consider comparing results of both studies: are they consistent?

4/- In light again of my first comment, there is currently a debate about the need of having interactive chemistry in climate model or if it is sufficient to prescribe chemistry. The concern is real given the heavy computational costs that interactive chemistry requires. This question could have been addressed here by using the chemistry outputs of the CHEM experiments as a chemistry forcing for a say "prescribed-CHEM" experiment with solar-induced ozone changes. Both effects (1)+(2) (see comment 2/) could thus have been considered without including interactive chemistry. Did the authors perform such experiments? If they have (and only if they have), it would be relevant to mention their conclusions in the paper.

**Specific comments**:

+ P1L6-10: "*In simulations with chemistry-climate interactions a second, dynamical effect on the AMOC is identified which counteracts the thermal effect. This dynamical mechanism is driven by the stratospheric cooling in response to the reduced solar forcing, which is strongest in the tropics and leads to a weakening of the Northern polar vortex. In simulations with interactive chemistry, these stratospheric changes are strongly amplified by the reduction of stratospheric ozone.*" The point made in these three sentences seems confusing. The first two sentences seem to suggest that the stratospheric cooling is found only in the chemistry-climate simulations while it is in fact found in both but amplified when ozone reduction feedback is included (as suggested by the third sentence) in addition to the direct radiative heating reduction. This may benefit of being clarified.

+ P2L12-13: "*The variability of the overturning circulation is furthermore influenced by external forcings (Otterå et al., 2010). Volcanic eruptions have been found to intensify the AMOC on decadal time scales (Otterå et al., 2010; Mignot et al., 2011).*" Since the study particularly investigates the mechanisms, I would suggest here to specify through which mechanisms volcanic eruptions influence AMOC (i.e. direct radiative cooling effect + tendency to induce positive NAO).

+ P2L21: change "*trough*" to "through"

+ P2L34-P3L1: "*This response is modulated by chemistry-climate interactions. In particular, stratospheric ozone reacts to the UV changes and amplifies the stratospheric temperature change (Baldwin and Dunkerton, 2005)*". I think that further explanations on the UV-ozone-temperature interactions may be needed given that they are the source of the difference found

between the CHEM and NOCHEM versions of a same experimental scenario. Furthermore, the reference to Baldwin and Dunkerton (2005) might not be the best suited for this purpose. The authors could rather refer to the work of J. Haigh in the 1990s (Haigh, 1994 ; 1996). The authors could also refer to section 3.5 of the CCMVal report (and reference therein) which can be found at the following address http://www.sparc-climate.org/publications/sparc-reports/sparc-report-no5/. This chapter particularly details the implication that prescribing constant ozone (as in the NOCHEM experiments of the present study) has on shortwave heating rates associated with changes in the UV.

+ Section "*2.1 The model*": What about energetic particle effect? SOCOL-MPIOM has parameterizations that allow taking into account GCR and EPP effects (which are linked to solar activity variations) and are suggested to also have an impact on the Northern Hemisphere surface climate (e.g. Rozanov et al. (2012)) through the "top-down" mechanism and thus may also affect the AMOC.

+ P5L12-20: Here the authors may consider discussing their results in comparison with Chiodo and Polvani (2016).

+ P6L4-5: The sea-ice extension and the associated differences between S2-CHEM and S2-NOCHEM experiments are hard to see on Fig 2 which is already quite busy.

+ P6L13-14: "*Additionally, a significant reduction of the precipitation is found in the North Atlantic, which further increases the salinity.*" Please indicate that this is not shown (in brackets).

+ P6-7: "*3.1 The thermal effect of SRR on the AMOC*": This part contains very interesting material and is very informative. However, I found quite hard to follow the text and figures together. While this is largely due to the fact that I am not used to examine ocean processes, I believe that some improvements could still me made. In particular, one of the key points relies on the differences, between the CHEM and NOCHEM configurations, of the timing of the anomalies development leading to differences in the AMOC response. In this regard, I think that, in addition to spatial patterns (Fig. 3), showing time series (similar to Fig. 1) of the key variables in the key regions may help understanding the timing issue.

+ P7L19-20: "*For S2_CHEM, a pronounced weakening of both polar vortices is found.*" Please give the reference to Figs. 4d,e,f in the text and replace "*both polar vortices*" by "NH and SH polar vortices" for clarity concerns.

+ P7L25: Is it annual anomalies or only winter (NDJFM) anomalies which are shown in Figs 4 and S3? Please clarify.

+ P7L26-27: Again for clarity, one sentence to explain what a SSW is may be useful here.

+ P8L3-4: "*Overall, the downward coupling of wind speed anomalies does not differ substantially between the CHEM and NOCHEM control experiments.*" Although it is written that the statement concerns "*anomalies*", I believe that this sentence might be misleading since it seems to suggest that the CHEM and NOCHEM downward influence of the stratosphere on the

troposphere are the same. We thus may wonder why we should expect a difference in the AO strength (described in paragraph which follows, P8L5-14). Please make this point clearer (as it is a key point of this paper).

References:

- Chiodo, G., and L. M. Polvani (2016): Reduction of climate sensitivity to solar forcing due to stratospheric ozone feedback, *J. Clim.*, Doi:10.1175/JCLI-D-15-0721.1.
- Cionni, I., V. Eyring, J.F. Lamarque, W.J. Randel, D.S. Stevenson, F. Wu, G.E. Bodeker, T.G. Shepherd, D.T. Shindell, and D.W. Waugh (2011): Ozone database in support of CMIP5 simulations: Results and corresponding radiative forcing. *Atmos. Chem. Phys.*, 11, 11267-11292, doi:10.5194/acp-11-11267-2011.
- Nowack, P.J., et al., (2015), A large ozone-circulation feedback and its implications for global warming assessments, *Nat. Clim. Change.*, 4, 41-45, doi:10.1038/nclimate2451.
- Rozanov, E., et al. (2012), Influence of the Precipitating Energetic Particles on Atmospheric Chemistry and Climate, *Surv. Geophys.*, 33:483-501, doi:10.1007/s10712-9192-0.
- Son, S.-W., et al. (2008), The impact of Stratospheric Ozone Recovery on the Southern Hemisphere Westerly Jet, *Science*, Vol 320, Issue 5882, doi:10.1126/science.1155939.

---

## Referee Comment (RC3) · Anonymous Referee #3 · 28 May 2016

The paper by Muthers and coauthors assesses the potential impact of atmospheric chemistry on the Atlantic meridional overturning circulation (AMOC) in two scenarios of reduced solar incoming radiation. The analysis is performed in ensembles of simulations in which interactive atmospheric chemistry is switched on and off. This allows the authors to detect two competing mechanisms that act toward strengthening and weakening the AMOC: the former as a result of thermally driven changes in upper ocean densities; the latter as a response of a dominating Arctic Oscillation negative phase, which in turn results from changes in the stratospheric circulation. Muthers et al. therefore conclude that the inclusion of atmospheric chemistry in climate models could be essential for a correct representation of solar-driven AMOC changes. These results could be of great relevance for the community and, hence, worth publicing. However, my main concern about this paper relates the fact that the Introduction, as it

is written now, does not allow us to to clearly see the novelty behind this investigation, or whether this is relevant at all. The Introduction lacks a clear description – which, on the the other hand, does not have to be too long – of previous works on the same or similar fields, so that we can identify from the very beginning what is the "hole in our current knowledge" the authors aim to address. I must admit that this is partly done in the last paragraphs in the Conclusion section; however, it is here too late and must appear earlier in the paper. This task could actually be done at cost of the initial description of the AMOC, which is supplementary (my guess is that any one approaching this paper will already have a clear idea of what the AMOC looks like). The Introduction might thus be kept relatively short. I encourage the authors to revise the Introduction to clarify this aspect. For this reason, I recommend major revisions before considering this work for publications

Other major points

The experiments A small comment of why control simulations where simulated under 1600 CE conditions is recommendable, as CMIP5, for example, suggested using 1850 CE conditions. Also, why were the simulations run only 30 years? Is there any particular reason?

Results Could the authors also show the pattern of AMOC anomalies as a result of reduced incoming solar radiation? I think an index alone is not sufficient, and AMOC anomalies might be of different signs on different sites. This might indeed be interesting to show and comment.

Discussion Discussion might be enriched by putting this work's results into the context, for example, of some solar minima in the recent past, like the Maunder Minimum. Also, it might be interesting to discuss the changes one might expect if solar variability changes were indeed of smaller magnitude, as some reconstruction suggest. Would the authors expect a similar response in the AMOC/climate?

Minor Comments

Page 1 L4. SRR acronyms is not used in Abstract L18 . . . upwelling processes that bring the water back . . . L19 please, rephrase "this Atlantic circulation" L20 I think, there is no need to bring the Atlantic Meridional Oscillation into the discussion if this is not going to be used any further

Page 2 L4-5 Upper salinity also increases due to net evaporation in the tropical North Atlantic L22 Please, remove comma after management L23 GHG has not been defined

Page 3 L5 "different mechanisms, how" please, rephrase L9 add comma after chemistry

Page 4 L27 Do experiments here mean simulations? I suggest reviewing the use of these two terms throughout the manuscript, as sometimes one feels they are interchanged. L32-33 there is no need to indicate that AO index is multiplied by -1

Page 5 L2 "near-surface (2 m) air temperature" L2-end I wonder why common acronyms are not used throughout the text, such as, SAT, SST, etc. L2-end In many instances it is written: "reduction in temperatures". This can be perfectly replaced by "cooling" L7-11 This is a topic for the Discussion. It is nonetheless of little relevance for this paper. L18 "are related"

Page 6 L4-5 It is not clear in which run the larger cooling is found L5-6 do temperatures and sea ice anomalies here refer to the value or the pattern? Please, clarify. Besides, it is said that they are similar, but not to what. Does it mean similar to those in S2? L12 add comma after "sea ice formation" L15 Here I wonder how relevant it is for the sea ice increase the advective contribution from a stronger AMOC. L17 Replace everywhere in the text Nordic Sea for Nordic Seas, as it stands for Greenland, Norwegian, Iceland seas, and sometime also the Barents Sea. L23 please, rephrase ". . . but the significance is reduced" L24 add comma before while L28 This sentence is probably too long. It could be divided into two. L30 please, clarify or rephrase "in other parts of the North Atlantic" L30 remove comma after period L33 remove comma after convection L35 rephrase "Similar to the Nordic Seas" (for example, "As in the Nordic Seas,")

Page 7 L6 add comma after forcing L7 Split the sentence into two. "in comparison to S2_NOCHEM. Similar differences ... " L11 This statement might need a citation L17 add comm after forcing L21 add comma before a reduction L23 add comma after Furthermore L25 It is interesting to notice that changes in the polar vortex do not seem to go linearly with the reduction in the solar forcing. One should not expect linearity in the response, of course, but it is interesting in any case.

Page 8 L9 add comma after response; change phenomena for phenomenon L10 add comma after AO index L12-14 I do not necessarily agree with the authors on some of the interpretations they make from Figure 6 regarding the AO index, which are in these lines exposed. For example, changes in the S1 experiments are mostly nonsignificant, and, although in CHEM there is a shift toward more negative values, in NOCHEM the change is more like a broadening of the distribution, rather than a change to more negative phases. Also, it should be stated here that the AO index in S2_NOCHEM features a first half of mostly negative values, followed by a positive trend towards more positive. This might even be investigated further, as an extra. L16 affects L25 Here a statement connecting changes in temperature and salinity with those in density might be help connect ideas. L27 Could you explain shortly or cite in the literature why this instantaneous AMOC response to the AO? Is it due to wind forcing? If it were due to heat-driven changes in the convection, as those found during positive or negative phases of the NAO, I would assume some delay in the response of the AMOC L33 Add comma after As a consequence, L34-35 Isn't it a reduction in the density? Otherwise, one should not expect a reduction in the convection, but an intensification

Page 9 Conclusions: I'd call this section Conclusions and Discussion. L6 please, remove comma after chemistry L12 the sentence about the projected future weakening of the AMOC should be connected with the next paragraph. L15 It would be recommendable to compare the magnitude of the projected minimum with that of those implemented in this study, as well as its duration. If the magnitude of this future minimum were much smaller, we might then expect negligible changes in the AMOC strength

[Figure]

L20 please, rephrase. For example, adding after effect "when atmospheric chemistry is taken into account" L25 Many of the elements? L26 on various time scales. Also, it would be recommendable to indicate which scales in particular the authors refer here L25-30 In this paragraph, three different verb tenses are used to talk about results from previous studies. I suggest using only one, maybe past simple? L31 remove comma after for the first time

Page 10 L2 when chemistry-climate interactions. . . this, I think, is already indicate at the beginning of the sentence L4 remove comma after GHGs L7 add comma after In this case,

FIGURES Would it be recommendable to add some of the Supplementary Figures to the main text? In particular those that are most referred in the text. There are indeed more Supplementary Figures that main ones. Fig. 1 Please, clarify whether the Student's t-test done after or before smoothing? The gray vertical lines indicating the SRR period are black Fig. 2. Please, clarify why climatologies in panels e and g, and in f and h are different, if they derive from the same control simulation, CHEM and NOCHEM respectively? Figs. 5 and 6 Gray vertical lines are again black Fig. 7 Readjust text to match the panels Fig. 8 Could you please increase the font size of the smallest text? Fig. S4. What are the shading and contours respectively?

---

## Referee Comment (RC4) · Anonymous Referee #4 · 30 May 2016

————————— General comments —————————

This paper is presenting different sets of simulations that evaluate the impact of a decrease in Total Solar Irradiance (TSI) over three decades, with a specific attention to the AMOC. It is focusing on the impact chemical changes induced by such a decrease, through comparison of a model not including this process, and another one including it. In both models, the decrease of TSI leads to an AMOC strengthening in the decades following the onset of the decreased TSI. The authors argue that this strengthening is larger when the chemical processes are not accounted for. They attribute such an effect to the impact of stratospheric chemistry has on the AO response to TSI decrease. Indeed, TSI decrease may lead a negative NAO due to larger cooling in the stratosphere associated with ozone depletion, which when reaching the surface may affect air-sea fluxes and wind stress, decreasing in particular salinity, which may diminish

salinity in the ocean convection sites, limiting AMOC enhancement.

As the former summary shows it, the amount of results shown in this paper is very significant. The topic is also of large interest, since the climatic impact associated with AMOC is well known as well as its good predictability a few decades ahead, and the TSI is also potentially largely predictable and is believed to decrease substantially in the coming decades. The impact of chemistry in the stratosphere was believed to potentially impact the AMOC response to TSI (e.g. Ottera et al. 2011), and this is the first study I see that tackle this potentially important process.

The paper is generally correctly presented, even though I have a large number of comments to clarify and better present the results. My main concerns are that:

1. the main effect analysed (i.e. the impact of chemistry on AMOC response to TSI decrease) is very small and maybe hardly significant;

2. the demonstrations are sometimes too rapid;

3. the amount of nice results is maybe too large, which may request to separate the analysis into two papers, i.e. two parts of the main analysis. The first dedicated to a better understanding of AO/NAO response, which is already largely depicted in the present paper, and constitute a very important results, even if not new. The second one will be dedicated to the analysed of the AMOC, which deserves a few more analysis, especially since it is the main topic of the present paper, but only have a few figures that are directly analysing the process involved in the presented changes.

Concerning the impact of the AMOC, I'm not entirely sure that the effect of chemistry leads to significant results. The ensemble mean of the simulation seems a bit different, but no error bar, nor statistical test are applied to confirm the supposed impact. Generally speaking, the differences between the two sets should be more systematically highlighted as in Fig. 4 (right panels), which is not the case everywhere, as well as the error bar associated with ensemble spread. Since this is the main result highlighted in

the paper, this should be proven with more statistical confidence, or the main message of the paper should be modified.

For all these reasons, although I found the set of experiments very interesting and potentially improving our understanding of climate dynamics in response to solar forcing, I found the take-home message and general descriptions of the results and logical connections sometimes a bit rapid. I therefore consider that the manuscript need major revisions before to be published, and I will advise the authors to consider the possibility of splitting their results into two parts (and two papers) in order to describe properly the main results and mechanisms discussed.

————————- Specific comments: ————————-

- P. 1, l. 20: "is responsible for the temperature conditions in western Europe": there is a lot of debate on this specific topic: cf. Seager et al. (2002). The AMOC does not have only an impact on western Europe and cannot explain the whole climate of this region

- P. 1, l. 22: "Meehl et al. 2009b": 2009a shoud come first.

- P. 1, l. 23: add "in the past" after climatic changes"

- P. 2: l. 13: "eruptions have been found to intensify the AMOC on decadal time scales": this is not just a question of intensification, but rather of variability excitation cf. Swingedouw et al. (2015)

- P. 4, l. 30: "monthly mean": This is a surprising choice. By doing so you include large part of so called Ekman wind-driven variability. Have you tried to remove this component, or to consider annual mean to limit its influence.

- P.5, l. 3: what are the spread or error bar associated to the value given (since we are here considered ensemble of simulations.

- P. 5, l. 8: can you be more specific on the reference that gave the climate sensitivity
of the model and the computation you have made. When you gave numbers, you have to be more specific on the way you compute them.

- P. 5, l. 19-20: why is outgoing longwave increasing when water vapour increases. Please clarify the process at play here.

- P. 6, l. 1: why don't you look at the NAO rather than the AO, since you are looking at the North Atlantic region. The two are usually very much linked, but can you confirm this in your model?

- P. 6, l. 9: "the sea ice differences": when? Try to be very precise on what you are talking about

- P. 6, l. 9: "therefore": the logical connection is not very clear to me, please clarify it.

- P. 6, l. 17: "Nordic Sea". I am usually seeing "Nordic Seas", since it is a few seas that you are dealing with (Greenland, Iceland, Norway). The same elsewhere in the ms.

- P. 7, l. 8-9: I'm not convinced the anomalies between CHEM and NOCHEM are significant for the AMOC. Please provide appropriate statistics.

- P. 7, l. 11: can you provide a reference or an explanation to support this claim?

- P. 7, l. 13: "28K" is this concerning only a grid points?

- P. 7, l. 20: "-43%": when? Over the 30-year period?

- P. 7, l. 23: what is your definition for the "duration of the winter period"?

- P. 7, l. 24-25: a series of number are given, with very poor definition. Please clarify.

- P. 8, l. 3: "downward coupling": can you define this?

- P. 8, l. 21: "freshwater flux": from which component? Precipitation? Evaporation? Sea ice?

- P. 8, l. 23: "export of saline water from the Nordic Sea by EGC". The EGC is a very

fresh and cold current, so it is not exporting saline water! Do you mean the weakening of this current is increasing the salinity? Please clarify.

- P. 8, l. 28: "instantaneously": thus, this is likely not related to convection but rather to wind-driven changes. Can you comment on that?

- P. 8, l. 31: "weaker intensification": significant? At which level? (please account for autocorrelation when computing degrees of freedom, since the AMOC has very low variability.

- P. 9, l. 11: "is also one of the": not really, since in projections, this is the longwave radiation that is mainly affected rather than the solar radiation changes.

- P. 9, l. 20-24: while the impact on the AO is very large, the impact on the AMOC is very weak, why is that? Is it coherent with small effect of AO on AMOC in control? What is the regression value of the AO on the AMOC in this model? Lohman et al. (2009) can be an interesting references concerning long term of a positive NAO on North Atlantic.

- P. 9, l. 32: "importance": I think this is a strong statement for a very weak effect in the end. . .

- P. 10, l. 1: add "slightly" after "may"

- Fig. 1: please compute a statistical test for differences between CHEM and NOCHEM anomalies.

- Figs 2,3, 7: please compute the difference CHEM-NOCHEM as in Fig. 4

- Fig. 7: This is a key figure when trying to understand what is going on for the AMOC, which should be the heart of the paper, given the title. Why is the projection so different than in 3? We want to see what is going in the whole Nordic Seas, including Fram Strait. What about circulation changes (barotropic stream function for instance)? wind stress? Density? Thermal and salinity component of density? The demonstration of

the processes affecting the North Atlantic should be more depicted. Figure S2 in this regard is interesting and should come in the main ms., but what is missing on this figure is an indication of the time frame. When are the changes occurring. Each point corresponds to a year from what I understand (with a smoothing of 15 years). Thus the anomalies are firstly thermally driven and then salinity driven. Why is there such a 10-year lag? (which is not clear from Fig. 8 where no time scale is shown).

——————- Bibliography: ——————-

- Lohmann K, H Drange, M Bentsen (2009) Response of the North Atlantic subpolar gyre to persistent North Atlantic oscillation like forcing. Climate dynamics 32 (2) pp 273-285

- Seager R, DS Battisti, J Yin, N Gordon, N Naik, AC Clement and MA Cane (2002) Is the Gulf Stream responsible for Europe's mild winters? QJRMS 128 (5), pp. 2563-2586

- Swingedouw D, P Ortega, J Mignot, E Guilyardi, V Masson-Delmotte, PG Butler and M Khodri (2015) Bidecadal North Atlantic ocean circulation variability controlled by timing of volcanic eruptions.Nature Communications 6, pages: 6545

---

## Author Comment (AC1) · 6 Jul 2016

We would like to thank referee 1 for his/her constructive and detailed review.

This is a mostly well written manuscript with interesting new results identifying a stratospheric mechanism impacting the Atlantic Meridional Oscillation. Therefore, it is potentially suitable for publication in the Earth System Dynamics journal. I have, however, a few concerns which I would like the authors to address before I can recommend the publication.

**My major concerns are:**

1. What would be the impact of aerosols? Your model does not include aerosol interactions, you just simply reduce the solar radiation. This seems a critical simplification to me. You should at least discuss how aerosol interactions would modify the AMOC response if taken into account in your model.

We agree, that the response of the coupled atmosphere-chemistry-ocean model to stratospheric aerosols is the next follow up question, which should be addressed. In this study, however, we focus on response of the system to a direct reduction in the solar energy input (i.e. total solar irradiance). A reduction of the TSI takes place during grand solar minima (e.g., Dalton Minimum) or in the case of solar radiation management techniques taking place in space (e.g., reduction of the TSI by mirrors in space). The model includes also aerosol interactions and indeed a number of modelling studies on the response to stratospheric aerosols from volcanic eruption have been performed earlier (e.g., Anet et al, 2013, Muthers et al. 2014 and 2015).

A comparison between both approaches, a reduction of the TSI space and through stratospheric aerosols is, however, highly relevant. We therefore discuss possible effects of radiation management by stratospheric aerosols at the end of the submitted manuscript.

*"The dynamical effect is expected to change, however, when the solar radiation is reduced in the Earth's atmosphere, for instance, by stratospheric sulphate aerosols. In this case a strengthening of the NH polar vortex and a positive phase of the AO may develop, analogous to the response to strong tropical volcanic eruptions (Graf et al., 1993; Kodera, 1994; Stenchikov et al., 2002; Muthers et al., 2014a, 2015). This effect of the positive AO phase may, in turn, lead to an intensification of the AMOC. Future studies shall address the influence of stratospheric sulphate geoengineering on the AMOC and the possible role of chemistry-climate interactions."*

2. I think in reality the salt rejection from the sea-ice growth is rather small and mainly occurs north from the regions of deep convection. Therefore, it has only a minor importance to the deep convection and the AMOC compared to the heat loss and possibly the net precipitation (precipitation minus evaporation) at the ocean surface. At the moment, the reader is led to understand that the salt rejection is at least as important as the heat loss. The increase of the sea-surface salinity could also be due to a decreased net precipitation related to changing storm tracks, for example. To better support the salt rejection argument, you need to quantify the salt rejection to the surface density and compare it to other factors. Can you check the ocean surface fluxes from your model output and their relation to the T and S, not only density, anomalies? How realistic these modelled fluxes then

are, depend on your model skill and are related to your model configuration, such as the sea-ice salinity scheme.

Thank you for this comment. We will address the importance of salt rejection in the revised manuscript. Probably, our statements were a bit too strong at some points and may require clarification.

3. I have a problem when you treat the AO and NAO identically. Although the AO and NAO correlate, they are not identical, not even from the AMOC perspective. I agree that the AO behaves largely like the NAO in winter. If, instead of the AO, you based your analysis on the NAO, how would the results look like? What would be their significance after taking into account the possible year-to-year autocorrelation?

See in comments below.

**Minor comments:**

- Page 1, line 16. I would rather say that 'surface currents transport water into the northern North Atlantic' rather than to 'Northern high latitudes' which sounds more like to the Arctic Ocean.

We changed 'Northern high latitudes' to 'North Atlantic'.

- Page 2, line 7. I don't think the AO is the hemispheric equivalent of the NAO. The NAO is a regional index and correlates with the AO, but their definitions differ substantially.

We have deleted the phrasing 'hemispheric equivalent'

- Page 2, lines 18. '... by increasing SSTs and enhancing freshwater input ...'

Rewritten to: "An increase in the solar forcing has been found to weaken the AMOC by increasing SSTs and enhancing freshwater input (Cubasch et al., 1997; Latif et al., 2009; Otterå et al., 2010; Swingedouw et al., 2011)"

- Page 2, line 32. As you focus on the AO in this paper, would be clearer not to talk about the NAO, but the AO, after Page 2, line 7.

We agree and focus on the AO in the revised manuscript. Note, that the results are very similar, when the analysis is performed using the NAO index.

- Page 3, line 19. '... uses temperature data ...'

Done.

- Page 4. line 3. You provide very little details on the model configuration. For example, what was the time step you used? How about the sea-ice salinity, was it constant? Or what sea-ice thermodynamics scheme was deployed? This information is important to assess how realistically the sea-ice salt rejection was modelled.

The model used is a configuration of the widely used ECHAM5-MPIOM (COSMOS model), which has been applied in various modelling studies and the IPCC AR4. The only difference between our version and the COSMOS version is the coupled atmospheric-chemistry module and this configuration is described in great detail in Muthers et al. (2014b).

The requested information has been included in the model description of the revised manuscript:

- Sea ice thermodynamics: "Sea ice dynamics are based on the viscous-plastic rheology formulated by Hibler (1979)."

- "A constant sea ice salinity of 5 psu is assumed."

- Time-step: ""The time-step of the atmospheric component is 15 minutes, with the radiation and the chemical computations performed every 2 hours." Ocean: "The time-step of the oceanic component is 2 hours and 24 minutes."

- Page 4, line 20. You should mention here how long model simulations continued after the 30 year SSR period.

Rewritten to: "The reduction of the solar forcing is switched on in year 5 of a simulation and lasts for 30 years when it is switched off and the simulation is continued for 25 years."

- Page 4, line 27. Explain the acronym TSI.

The acronym TSI is now defined at its first occurrence.

- Page 4, line 32. Explain more in detail how the AO index was calculated and provide references. For example, a common way to calculate the AO is based on the PC1 of 1000 mb pressure height anomaly data north of 20N. Your method seems to differ from that. Why? How robust your results are based on the AO calculation method?

We compared both AO methodologies, the EOF based way and the simplified AO index using the sea level pressure (SLP) north of 70°N. Both indices are closely related, for the CHEM_CTRL we find a Pearson correlation coefficient of 0.81 (0.85 for NOCHEM_CTRL). When calculating the AO based on sea level pressure data a common approach is based on the difference in the zonal mean SLP between around 40°N and 65°N (e.g., Li and Wang (2003), Braesicke and Pyle (2004) ). We also compared our index (SLP field north of 70°N) to the index using the SLP difference between 40°N and 65°N and found very similar results. We therefore conclude that the exact definition of the AO index is not important for the results of this study.

However, for a better agreement with previous studies we will think about applying an AO index based on the SLP difference between 40°N and 65°N in the revised manuscript. In comparison to the EOF based definition we prefer this approach for its simplicity. In this case we will update the result and figures in the revised manuscript.

- Page 5, line 18. '... are related ...'

Done.

- Page 5, lines 21-22. This sentence is hard to understand. How is the slight initial reduction of the global mean temperature related to the initial conditions of the ocean when the ocean initial conditions are from a 1300 year long simulation? Why rather not related to the atmospheric initial conditions which presumably started from an observation based, physically less consistent initial state?

The oceanic restart file is identical in all ensemble simulations. The atmosphere is perturbed by slight time difference between the restart files. Therefore, there is a considerable amount of "memory" in the ocean, which dominates the behaviour of the AMOC during the first years. We rewrote the relevant sentences for the revised manuscript:

"A slight initial reduction of the global mean temperature is also found in the reference ensemble experiments and is related to the initial conditions of the ocean. With all ensemble simulations sharing the same oceanic conditions in the beginning, the AMOC evolution of the first years is dominated by the oceanic memory."

- Page 5, line 31. 'during the second half'

Thank you.

- Page 6, line 7. Do you mean that sea-ice patterns look similar but their anomalies are (presumably) weaker in the S1 experiments?

Exactly. Rewritten to: "In the S1 experiments similar but weaker temperature and sea ice anomalies are found and S1_CHEM experiment is characterized by an amplified temperature reduction as well (not shown)."

- Page 6, line 13. Is the reduction in precipitation related to a shift in the main storm track and, as a result, a colder and dryer atmosphere?

It is mainly due to the shift in the storm track. The negative anomalies in the North Atlantic and Northern Europe occur along with positive anomalies in Southern Europe (compare Fig. R1).

[Figure]

*R 1: Precipitation anomalies in S2_CHEM relative to CTRL_CHEM averaged over the 30 year SRR period*

We changed this sentence in the manuscript accordingly: "Additionally, a shift of the storm track and a significant reduction of precipitation is found in the North Atlantic, which further increases the salinity."

- Page 6, lines 15-16. I think you need to verify the significance of the salt rejection to the surface density. It is typically small compared to the cooling effect. Also, not much freezing occurs at the

eastern side of Greenland, but the Arctic ice flows south and melts along the eastern boundary of the East Greenland Current.

See above. We will address the issue of salt rejection and transport in detail when revising the manuscript. However, based on Figure S2 in the submitted manuscript, we are confident, that salinity (either through salt rejection, transport or reduced precipitation also play a role in the density changed.)

- Page 6, lines 26-27. You should mention that these density and mixed layer depth anomalies are not reflected in the AMOC.

We have added: "These changes during the first 15 years are not reflected in the AMOC index."

- Page 6, line 30. '... the North Atlantic (Fig. 3a).'

Done.

- Page 6, lines 33-34. You must mean 'the central North Atlantic' here.

Thank you.

- Page 7, line 2. The 'dominance' is based on very speculative assumptions. Just say 'Salinity changes, nevertheless ...'

Changed accordingly.

- Page 8, line 1. Add a literature reference that proofs the linkage between the downward propagating wind anomalies and the AO phases.

At this point in the manuscript we discuss Fig. S4, which shows a connection between the AO and zonal mean wind anomalies. The pattern of the zonal wind composite suggests downward propagation of the wind anomalies. A similar pattern has been found by Baldwin and Thompson (2009, compare their Fig. 11) and we have added this reference to the manuscript.

- Page 8, lines 13-14. You don't show this in Fig. 6, which should be mentioned, or plot CTRL_NOCHEM in Fig. 6.

We show boxplots of the AO index for all experiments in Fig. 6 c and d these show the behaviour we discuss in the manuscript. In the revised manuscript we have added an explicit reference to Fig. 6 d to the corresponding explanation.

- Page 8, line 16. '... which affects the wind ...'

Done.

- Page 8, line 29. '... the AO phase has a long lasting effect ...'

Done.

- Page 8, line 33. This should be '... the weakening of the Northern polar vortex ..', right?

Right, thank you.

- Page 8, lines 34-35. I suggest you to write '... dynamical changes decrease the density of the surface ocean waters South of Greenland, ...'

Applied as suggested.

- Page 9, lines 11-13. Don't these citations analyse the impact of the increase in GHGCs? Seems like you are cutting corners here. Wouldn't it be more correct to say e.g. '... Swingedouw et al., 2011). Related to increasing global greenhouse gas concentrations and associated surface warming, it is also one of the dominant ...'

We agree that our explanation was a bit oversimplified. We changed this as suggested to: "This response of the overturning to solar radiation changes has been identified in earlier studies (Cubasch et al., 1997; Latif et al., 2009; Otterå et al., 2010; Swingedouw et al., 2011). Related to increasing global greenhouse gas concentrations and associated surface warming, it is also one of the dominant mechanisms for the projected future weakening of the AMOC (Stocker and Schmittner, 1997; Manabe and Stouffer, 1999; Mikolajewicz and Voss, 2000; Gregory et al., 2005; Stocker et al., 2013)."

- Page 9, line 17. '... may reduce the projected 21st century ...'

Done.

- Page 9, line 18. '... stronger than in the late 21st century than [today?], when a grand ...'

We rewrote this to: "This is confirmed by simulations of Anet et al. (2013b). The AMOC is significantly stronger in the late 21th century, in an experiment including a grand solar minimum in the second half of the 21th century in comparison to simulations without a decline of the solar activity (Fig. S7)."

- Page 9, line 23. '... the AMOC by anomalous …

Done.

- Page 9, lines 23-25. This sentence is not clear to me. I suggest rewriting 'The dynamical effect is enabled by chemistry climate interactions, which result in amplified stratospheric temperature responses.'

We rewrote this to: "The dynamical effect is amplified by chemistry climate interactions, which result in amplified stratospheric temperature responses." We prefer the term amplified, since the same dynamical effects is also present without chemistry-climate interactions, but much weaker.

- Page 9, lines 28-29. The literature you cite here include three studies analysing NAO and only one analysing AO. This indicates to me, that NAO would have been a more appropriate index for this study as well, although its relation to the polar vortex is not as clear as the one of the AO.

For the analysis of the stratosphere troposphere interactions the AO index is the more appropriate parameter. The AMOC index, however, is stronger influenced by the NAO. In our study, the influence of the stratosphere on the AMOC is analysed and therefore, we have to decide for one of the two indices to draw a consistent picture, from the stratosphere down to the ocean.

We argue, however, that the decision for one index or the other does not affect the conclusions of our study. Both indices are closely related in SOCOL-MPIOM. For CHEM_CTRL we find correlation coefficient of about 0.71 (0.68 in NOCHEM_CTRL.) For comparison we also performed our analysis with the NAO index (defined by the pressure difference between Iceland and the Azores). The results are given below and do not differ substantially between the results for the AO index.

[Figure]

*R 2: Similar to Fig 6 a/b of the submitted manuscript, but for the ensemble mean winter (Nov. – Mar.) NAO index. The NAO is defined by the sea level pressure difference between Iceland and the Azores. Blue lines correspond to the NOCHEM experiments, orange lines to the CHEM experiments. Solid lines resemble the S2 and dashed lines the S1 experiments. Dots indicate winters with significant differences to the CTRL ensemble (Student t-test p ≤ 0.05).*

[Figure]

*R 3: Similar to Fig 6 d/e of the submitted manuscript, but for the ensemble mean winter (Nov. – Mar.) NAO index. Blue boxplots correspond to the NOCHEM experiments, orange boxplots to the CHEM experiments.*

[Figure]

*R 4: Similar to Fig 7 of the submitted manuscript, but for the ensemble mean winter (Nov. – Mar.) NAO index.*

- Page 9, line 31. 'the modelled response of the AMOC ...'

done.

- Page 10, line 5. '... weakening of the AMOC with climatic ...'

done.

- Page 10, line 10. What do you mean by 'Future studies'. Be more explicit. Are you planning to do this work?

We are currently not planning to study these questions, since generating an appropriate aerosol forcing for SOCOL-MPIOM is a complicated tasks which requires simulations with an external aerosol microphysical model Therefore, this is sentence is meant as a general suggestion to the community and we would like to keep this sentence as is.

- Figure 1. Write out the TSI acronym in the figure caption. As you used t-test for significance, did you check the autocorrelation or did you just treat each year as an independent variable? If years correlate, it affects your significance estimates. Explain more in detail what you did.

The TSI acronym is defined in the caption of the revised manuscript.

Dots in Fig. 1 represent year, where the SSR ensemble (e.g., S2_CHEM, 10 simulations) differs significantly from the control ensemble (e.g., CTRL_CHEM, 10 simulations). We therefore do a comparison of two data sets with 10 values each against each other. There is no autocorrelation, since we do not include any temporal information and the 10 experiments can be considered to be independent.

- Figures 2-4, 7, S1, S3, S5. Dots are not dark grey, but black. Better to say 'Black dots denote non-significant ...'

We replaced the figures with new versions using a lighter grey colour.

- Figure 2. More correct to say 'The sea-level pressure contour interval is ...'

Done.

- Figure S2. Indicate latitude and longitude locations of these T & S profiles.

We did not use a latitude longitude box to calculate the T & S profiles. Instead the grid cells were selected by their averaged mixed layer depth. We state this in the caption of Fig. S2:

"The deep water formation regions cover all grid cells with an annual mean mixed layer depth ≥ 250 m in the corresponding ocean basins".

References:

- Anet, J. G., Muthers, S., Rozanov, E., Raible, C. C., Peter, T., Stenke, A., Shapiro, A. I., Beer, J., Steinhilber, F., Brönnimann, S., Arfeuille, F., Brugnara, Y., and Schmutz, W.: Forcing of stratospheric chemistry and dynamics during the Dalton Minimum, Atmos. Chem. Phys., 13, 10951-10967, doi:10.5194/acp-13-10951-2013, 2013.

- Baldwin, Mark P., and David W J Thompson. 2009. "A Critical Comparison of Stratosphere – Troposphere Coupling Indices." Quarterly Journal of the Royal Meteorological Society 1672: 1661–1672. doi:10.1002/qj.

- Braesicke P, Pyle JA. 2004. Sensitivity of dynamics and ozone to different representations of SSTs in the Unified Model. Q. J. R. Meteorol. Soc. 130: 2033–2045.

- Li J, Wang JXL. 2003. A modified zonal index and its physical sense. Geophys. Res. Lett. 30: 1632, DOI:10.1029/2003GL017441.

- Muthers, S., Arfeuille, F., Raible, C. C., and Rozanov, E. (2015): "The impacts of volcanic aerosol on stratospheric ozone and the Northern Hemisphere polar vortex: separating radiative-dynamical changes from direct effects due to enhanced aerosol heterogeneous chemistry". Atmos. Chem. Phys., 15, 11461-11476. 10.5194/acp-15-11461-2015

- Muthers, S., J. G. Anet, C. C. Raible, S. Broennimann, E. Rozanov, F. Arfeuille, T. Peter, A. I. Shapiro, J. Beer, F. Steinhilber, Y. Brugnara, W. Schmutz (2014): "Sensitivity of the winter warming pattern following tropical volcanic eruptions to the background ozone climatology", Journal of Geophysical Reseach, 199, 3, 1340-1355, 10.1002/2013JD020138.

- Muthers, S., Anet, J. G., Stenke, A., Raible, C. C., Rozanov, E., Brönnimann, S., Peter, T., Arfeuille, F. X., Shapiro, A. I., Beer, J., Steinhilber, F., Brugnara, Y., and Schmutz, W. (2014b): "The coupled atmosphere-chemistry-ocean model SOCOL-MPIOM", Geosci. Model Dev., 7, 2157–2179, doi:10.5194/gmd-7-2157-2014.

---

## Author Comment (AC2) · 6 Jul 2016

We would like to thank referee 1 for his/her constructive and detailed review.

This paper examines the physical processes responsible for the AMOC response to reduced solar radiation and assesses the importance of chemistry-climate in modulating this response. By comparing two sets of climate model experiments, with and without interactive chemistry, the study demonstrates that climate models which do not consider stratospheric -namely ozone -chemistry may overestimate the sensitivity of the AMOC to solar forcing since the "top-down" influence (stratospheric influence on tropospheric circulation) is underestimated.

In my opinion, this work constitutes a very nice contribution for the broad climate research readership as it demonstrates, using the specific example of the AMOC, the prominent and complex connections between the different components which drive climate variability (going from the stratosphere chemistry to the ocean circulation). The work is well-framed in the current literature. I found the paper mostly clear, well written and scientifically sound. I think however that some improvements and clarifications could be made before publication. Please find my main comments/suggestions below:

**Main comments/questions**:

1. It has already long been recognized that atmospheric chemistry interacts with dynamics and that its consideration in climate models is crucial to adequately simulate climate variability (e.g. influence of the ozone hole recovery on the SAM trends in CMIP3 simulations by Son et al. (2008)). As a consequence, historical and projection climate simulations in CMIP5 for models without interactive chemistry were designed by prescribing chemical fields that consider long-term trends (Cionni et al., 2011). For CMIP6, the ozone prescription fields should be even further improved. So I would say that the current question regarding chemistry-climate interactions is: do we really need interactive chemistry? or can it just be prescribed? The other question is then how to prescribe it in the most accurate way (see e.g. Nowack et al. (2015)).

In my opinion, given the frame, the results and the conclusion of the present study, I think that the introductory part of the paper should –at least partly –review the recent advances regarding chemistry climate interaction. A lot has been done already and should not be ignored.

Thank you, we will add a discussion of previous studies in chemistry-climate interactions in the revised manuscript.

2. In the light of my previous comment, I would suggest the authors to explain more thoroughly how the combined UV+ozone effects modulate the heating rates in the stratosphere which is the starting point of the stratospheric mechanism discussed in the paper. The thermal modulation of the stratosphere through UV variations comes from two main effects: (1) direct shortwave heating through incoming UV absorption by ozone ($\lambda \sim 200\text{-}300$ nm), (2) ozone change ($\lambda < 242$ nm) which also affect shortwave heating rates. Both effects count significantly. Basically, and if I understood correctly, their NOCHEM experiment account for effect (1) only while CHEM account

for effects (1) + (2). I think such clarifications are easy to make and necessary since they help understanding the basic difference between the two experimental configurations (at least regarding stratospheric ozone which is the major solar effect). In the present version of the paper too few information are given on UV-ozone-temperature interactions and their implication on experimental setting (e.g. P2L32-P3L1, P5L12-14).

We will improve the description of the UV+ozone effect in the revised manuscript.

3/-A very recent study by Chiodo and Polvani (2016) has just been released in *Journal of Climate* and deal with –somewhat -similar problematics. They performed simulations that also present some similarities with those perform in the present work. While both studies have their own relevance and focus on different aspects, they also nicely complement each other. The authors may consider comparing results of both studies: are they consistent?

Thank you. We have added a comparison to the results of Chiodo and Polvani to the discussion section of the revised manuscript:

"Recently, Chiodo and Polvani (2016) assessed the role of the interactive chemistry on the temperature and precipitation response to increasing SSI. They identified a reduced sensitivity with interactive chemistry due to the effect of the ozone increase on the short-wave radiation balance. Our results for a SSI reduction indicate a slightly larger temperature sensitivity with interactive chemistry owing to the effect of the stratospheric water vapour and ozone changes on the long-wave radiation balance. These differences may be attributed to model differences or differences in the response of the climate system to increasing and decreasing solar forcing. A possible effect of the differences in the atmospheric response on the AMOC is not discussed by Chiodo and Polvani (2016)."

4/-In light again of my first comment, there is currently a debate about the need of having interactive chemistry in climate model or if it is sufficient to prescribe chemistry. The concern is real given the heavy computational costs that interactive chemistry requires. This question could have been addressed here by using the chemistry outputs of the CHEM experiments as a chemistry forcing for a say "prescribed-CHEM" experiment with solar-induced ozone changes. Both effects (1)+(2) (see comment 2/) could thus have been considered without including interactive chemistry. Did the authors perform such experiments? If they have (and only if they have), it would be relevant to mention their conclusions in the paper.

We agree that this is a highly relevant question. Unfortunately, we did not perform these simulations.

**Specific comments**:

+ P1L6-10: "*In simulations with chemistry-climate interactions a second, dynamical effect on the AMOC is identified which counteracts the thermal effect. This dynamical mechanism is driven by the stratospheric cooling in response to the reduced solar forcing, which is strongest in the tropics and leads to a weakening of the Northern polar vortex. In simulations with interactive chemistry, these stratospheric changes are strongly amplified by the reduction of stratospheric ozone.*" The point made in these three sentences seems confusing. The first two sentences seem to suggest that the stratospheric cooling is found only in the chemistry-climate simulations while

it is in fact found in both but amplified when ozone reduction feedback is included (as suggested by the third sentence) in addition to the direct radiative heating reduction. This may benefit of being clarified.

The abstract will be rewritten.

+ P2L12-13: "*The variability of the overturning circulation is furthermore influenced by external forcings (Otterå et al., 2010). Volcanic eruptions have been found to intensify the AMOC on decadal time scales (Otterå et al., 2010; Mignot et al., 2011).*" Since the study particularly investigates the mechanisms, I would suggest here to specify through which mechanisms volcanic eruptions influence AMOC (i.e. direct radiative cooling effect + tendency to induce positive NAO).

Rewritten to "Volcanic eruptions have been found to intensify the AMOC on decadal time scales (Otterå et al., 2010; Mignot et al., 2011), through a reduction of the SSTs and a shift of the NAO towards a positive phase."

+ P2L21: change "*trough*" to "through"

Done.

+ P2L34-P3L1: "*This response is modulated by chemistry-climate interactions. In particular, stratospheric ozone reacts to the UV changes and amplifies the stratospheric temperature change (Baldwin and Dunkerton, 2005)*". I think that further explanations on the UV-ozone-temperature interactions may be needed given that they are the source of the difference found between the CHEM and NOCHEM versions of a same experimental scenario. Furthermore, the reference to Baldwin and Dunkerton (2005) might not be the best suited for this purpose. The authors could rather refer to the work of J. Haigh in the 1990s (Haigh, 1994 ; 1996). The authors could also refer to section 3.5 of the CCMVal report (and reference therein) which can be found at the following address http://www.sparc-climate.org/publications/sparc-reports/sparc-report-no5/. This chapter particularly details the implication that prescribing constant ozone (as in the NOCHEM experiments of the present study) has on shortwave heating rates associated with changes in the UV.

Thank you, we will consider this comment and improve the discussion of the UV+ozone+temperature interactions in the revised manuscript.

+ Section "*2.1 The model*": What about energetic particle effect? SOCOL-MPIOM has parameterizations that allow taking into account GCR and EPP effects (which are linked to solar activity variations) and are suggested to also have an impact on the Northern Hemisphere surface climate (e.g. Rozanov et al. (2012)) through the "top-down" mechanism and thus may also affect the AMOC.

SOCOL-MPIOM includes an EEP and GCR parametrization, but we concentrate on the effects of solar irradiance keeping the same EEP and GCR because they should not be changed in SRM case. Therefore, we think that these processes are not substantially relevant for our study and do not need to be mentioned in the model description.

+ P5L12-20: Here the authors may consider discussing their results in comparison with Chiodo and Polvani (2016).

We have included a comparison with the results of Chiodo and Polvani (2016) in the discussion section of the revised manuscript *see above).

+ P6L4-5: The sea-ice extension and the associated differences between S2-CHEM and S2-NOCHEM experiments are hard to see on Fig 2 which is already quite busy.

We agree, therefore the sea ice is shown again on Fig. 3. However, given the large number of figures we prefer not to add additional figures to the manuscript.

+ P6L13-14: "*Additionally, a significant reduction of the precipitation is found in the North Atlantic, which further increases the salinity*." Please indicate that this is not shown (in brackets).

Done.

+ P6-7: "*3.1 The thermal effect of SRR on the AMOC*": This part contains very interesting material and is very informative. However, I found quite hard to follow the text and figures together. While this is largely due to the fact that I am not used to examine ocean processes, I believe that some improvements could still me made. In particular, one of the key points relies on the differences, between the CHEM and NOCHEM configurations, of the timing of the anomalies development leading to differences in the AMOC response. In this regard, I think that, in addition to spatial patterns (Fig. 3), showing time series (similar to Fig. 1) of the key variables in the key regions may help understanding the timing issue.

We show time series of the mixed layer depth and upper ocean density for the two convective regions (GIN Sea, i.e., Nordic sea, and North Atlantic) below. However, we prefer not to include these time series in the revised manuscript. The time series are dominated by large variability and it is very hard to identify clear differences between S2_CHEM and S2_NOCHEM from these figures. The average over two 15 years periods removes a lot of the year to year variability and helps to identify the main differences between the two ensemble experiments.

[Figure]

*R 5: Anomaly maps (averaged over the solar minimum) and time series of the mixed layer depth and upper ocean density for the two convective regions (GIN Sea, i.e., Nordic sea, and North Atlantic). S2_CHEM is shown in panels a), c) and d). The results of S2_NOCHEM is shown in b), e), and f). Blue and magenta boxed in a) and b) denote the areas for the time series. The time series of the mixed layer depth and upper ocean density anomalies are normalized using the mean value and standard deviation of the corresponding control experiment.*

+ P7L19-20: "*For S2_CHEM, a pronounced weakening of both polar vortices is found.*" Please give the reference to Figs. 4d,e,f in the text and replace "*both polar vortices*" by "NH and SH polar vortices" for clarity concerns.

Done.

+ P7L25: Is it annual anomalies or only winter (NDJFM) anomalies which are shown in Figs 4 and S3? Please clarify.

Fig. 4 and S3 show the annual mean anomalies. This is stated in the caption of Fig. 4 in the revised manuscript.

+ P7L26-27: Again for clarity, one sentence to explain what a SSW is may be useful here.

We have added a short explanation on SSWs:

"The weakening of the NH polar vortex is closely related to the occurrence of sudden stratospheric warming (SSW) events (Fig. 5). SSWs are stratospheric extreme events, in which the westerly flow

during winter time is reversed and a strong warming in the polar stratosphere can be observed. SSW events in the NH are associated with a 'break down' of the polar vortex."

+ P8L3-4: "Overall, the downward coupling of wind speed anomalies does not differ substantially between the CHEM and NOCHEM control experiments." Although it is written that the statement concerns "anomalies", I believe that this sentence might be misleading since it seems to suggest that the CHEM and NOCHEM downward influence of the stratosphere on the troposphere are the same. We thus may wonder why we should expect a difference in the AO strength (described in paragraph which follows, P8L5-14). Please make this point clearer (as it is a key point of this paper).

We do not find large differences between the two control experiments, suggesting that the interactive chemistry has no large effect on the dynamics and the variability, when all external forcings are kept constant. Consequently, the influence of the stratosphere on the tropospheric AO is comparable with and without interactive chemistry. This has also been found in earlier studies with SOCOL-MPIOM (compare Muthers et al. 2014.).

However, this does not mean, that no differences is found when a changing external forcing is applied. In fact, we show in our results, that the interactive chemistry leads to a strong differences in the stratospheric temperature change to the reduced solar forcing, which causes a stronger weakening of the Northern polar vortex, which in turn leads to a clear difference in the response of the AO. This response is not related to differences in the stratosphere-troposphere coupling between both experiments, but to a differences in the stratospheric response.

**References:**

- Chiodo, G., and L. M. Polvani (2016): Reduction of climate sensitivity to solar forcing due to stratospheric ozone feedback, J. Clim., Doi:10.1175/JCLI-D-15-0721.1.

- Cionni, I., V. Eyring, J.F. Lamarque, W.J. Randel, D.S. Stevenson, F. Wu, G.E. Bodeker, T.G. Shepherd, D.T. Shindell, and D.W. Waugh (2011): Ozone database in support of CMIP5 simulations: Results and corresponding radiative forcing. Atmos. Chem. Phys., 11, 11267-11292, doi:10.5194/acp-11-11267-2011.

- Nowack, P.J., et al., (2015), A large ozone-circulation feedback and its implications for global warming assessments, Nat. Clim. Change., 4, 41-45, doi:10.1038/nclimate2451.

- Rozanov, E., et al. (2012), Influence of the Precipitating Energetic Particles on Atmospheric Chemistry and Climate, Surv. Geophys., 33:483-501, doi:10.1007/s10712-9192-0.

- Son, S.-W., et al. (2008), The impact of Stratospheric Ozone Recovery on the Southern Hemisphere Westerly Jet, Science, Vol 320, Issue 5882, doi:10.1126/science.1155939.

References

- Muthers, S., Anet, J. G., Stenke, A., Raible, C. C., Rozanov, E., Brönnimann, S., Peter, T., Arfeuille, F. X., Shapiro, A. I., Beer, J., Steinhilber, F., Brugnara, Y., and Schmutz, W. (2014b): "The coupled atmosphere-chemistry-ocean model SOCOL-MPIOM", Geosci. Model Dev., 7, 2157–2179, doi:10.5194/gmd-7-2157-2014.

---

## Author Comment (AC3) · 6 Jul 2016

We would like to thank referee 1 for his/her constructive and detailed review.

The paper by Muthers and coauthors assesses the potential impact of atmospheric chemistry on the Atlantic meridional overturning circulation (AMOC) in two scenarios of reduced solar incoming radiation. The analysis is performed in ensembles of simulations in which interactive atmospheric chemistry is switched on and off. This allows the authors to detect two competing mechanisms that act toward strengthening and weakening the AMOC: the former as a result of thermally driven changes in upper ocean densities; the latter as a response of a dominating Arctic Oscillation negative phase, which in turn results from changes in the stratospheric circulation. Muthers et al. therefore conclude that the inclusion of atmospheric chemistry in climate models could be essential for a correct representation of solar-driven AMOC changes. These results could be of great relevance for the community and, hence, worth publicating.

However, my main concern about this paper relates the fact that the Introduction, as it is written now, does not allow us to clearly see the novelty behind this investigation, or whether this is relevant at all. The Introduction lacks a clear description – which, on the other hand, does not have to be too long – of previous works on the same or similar fields, so that we can identify from the very beginning what is the "hole in our current knowledge" the authors aim to address. I must admit that this is partly done in the last paragraphs in the Conclusion section; however, it is here too late and must appear earlier in the paper. This task could actually be done at cost of the initial description of the AMOC, which is supplementary (my guess is that any one approaching this paper will already have a clear idea of what the AMOC looks like). The Introduction might thus be kept relatively short. I encourage the authors to revise the Introduction to clarify this aspect. For this reason, I recommend major revisions before considering this work for publications

We will rewrite the introduction in revised manuscript.

**Other major points**

The experiments: A small comment of why control simulations where simulated under 1600 CE conditions is recommendable, as CMIP5, for example, suggested using 1850 CE conditions. Also, why were the simulations run only 30 years? Is there any particular reason?

The Control experiment, which was used to initialize the ensemble, was part of a study, which focuses on transient climate simulations for the period 1600 to 2100. Therefore, a 1600 control experiment was performed to generate starting conditions for the transient experiments. These experiments are described in Anet et al. 2013, 2014, Muthers et al. 2014.

We have added the following description to the experiment section of the revised manuscript: "The year 1600 was chosen, since a stable long-term control simulation with SOCOL-MPIOM was available from previous studies (Anet et al., 2013a, 2014; Muthers et al., 2014b). Note, the differences in the climatic conditions between 1600 and the commonly used year 1850 are small and both represent a preindustrial climate state."

Results: Could the authors also show the pattern of AMOC anomalies as a result of reduced incoming solar radiation? I think an index alone is not sufficient, and AMOC anomalies might be of different signs on different sites. This might indeed be interesting to show and comment.

The pattern of AMOC anomalies is shown below and we will add this figure to the supplementary material of the revised manuscript. Furthermore, we have added a discussion of the results:

"The differences between the AMOC index for S2_CHEM and S2_NOCHEM are also reflected in the anomaly pattern of the AMOC (Fig. S2). Within the first 15 years the intensification of the circulation is weak. Positive anomalies are found between 40◦ N to 65◦ N and between the surface and a depth of 2800 m depth. During the first half of the reduction period the intensification is slightly larger in S2_CHEM. A pronounced strengthening of the circulation takes part in the second half of the reduction period. Positive anomalies cover all latitudes from the equator to 65◦N and most levels between the surface and 3000 m depth. In the second half of the reduction period, the intensification is more pronounced in S2_NOCHEM."

[Figure]

*R 6: Atlantic meridional overturning streamfunction anomalies (Sv) for S2_CHEM (a,b) and S2_chem (d,e) and the difference between the two experiments (CHEM-NOCHEM, c,f). Top row (a-c) displays anomalies for the first half of the solar reduction period; anomalies for the second half are shown in the bottom row (d-e).*

Discussion: Discussion might be enriched by putting this work's results into the context, for example, of some solar minima in the recent past, like the Maunder Minimum. Also, it might be interesting to discuss the changes one might expect if solar variability changes were indeed of smaller magnitude, as some reconstruction suggest. Would the authors expect a similar response in the AMOC/climate?

We will add a brief discussion of our results for recent solar minima in the revised manuscript.

**Minor Comments**

**Page 1**

L4. SRR acronyms is not used in Abstract L18 . . . upwelling processes that bring the water back . . .

Done.

L19 please, rephrase "this Atlantic circulation" L20 I think, there is no need to bring the Atlantic Meridional Oscillation into the discussion if this is not going to be used any further

Done. L19 rephrased to "the surface branch of the AMOC". The AMO is mentioned, since it has been suggested to be an important component for the multidecadal climate variations in the European region (e.g., Knight et al, 2006). Since other studies found a close relationship between the AMOC and the AMO we think mentioning this process highlight the relevance of studies on AMOC variability. Therefore, we prefer to keep this sentence.

**Page 2**

L4-5 Upper salinity also increases due to net evaporation in the tropical North Atlantic L22 Please, remove comma after management L23 GHG has not been defined

L4-5 rewritten to "Additionally, the salt content increases, through evaporation in the tropical regions and salt rejection during sea ice growth." Other modifications applied as suggested.

**Page 3**

L5 "different mechanisms, how" please, rephrase L9 add comma after chemistry

Rewritten to: "The purpose of this study is to assess the influence of a reduction of the solar forcing on the AMOC."

**Page 4**

L27 Do experiments here mean simulations? I suggest reviewing the use of these two terms throughout the manuscript, as sometimes one feels they are interchanged.

Thank you. Simulations were indeed meant here. We carefully checked the manuscript and use the terms experiment/simulations in a consistent way now.

L32-33 there is no need to indicate that AO index is multiplied by -1

In our approach it is. We define the AO index by the area averaged sea level pressure north of 70deg N. In this case, a negative anomaly to the long-term average corresponds to a positive phase of the AO. For clarity, we prefer to state explicitly the multiplication by -1.

**Page 5**

L2 "near-surface (2 m) air temperature" L2-end I wonder why common acronyms are not used throughout the text, such as, SAT, SST, etc. L2-end In many instances it is written: "reduction in temperatures". This can be perfectly replaced by "cooling" L18 "are related"

Changes applied as suggested. Acronyms are used for terms which occur multiple times in the manuscript. SST or SAT are not mentioned so often in the text. Moreover, we already use a number of different acronyms (CHEM, NOCHEM, CTRL, …) and introduction additional abbreviations would improve the readability.

L7-11 This is a topic for the Discussion. It is nonetheless of little relevance for this paper.

We agree, that the relevance for the temperature differences between CHEM and NOCHEM and their relations to the models climate sensitivity are not very relevant the AMOC. However, it is relevant to understand the influence of the chemistry on the surface temperature variations and for the comparison of our results to earlier studies. Based on feedback from reviewer 2, we have

included a comparison of our results to the study of Chiodo and Polvani (2016), who analysed the role of chemistry-climate-interaction on the temperature response to solar forcing. Therefore, we would like to keep this brief description of the temperature signals and the comparison with the climate sensitivity.

**Page 6**

L4-5 It is not clear in which run the larger cooling is found

Changed to: " Furthermore, a larger cooling over the Barents Sea is found in S2_CHEM, which extends towards Northern Eurasia."

L5-6 do temperatures and sea ice anomalies here refer to the value or the pattern? Please, clarify. Besides, it is said that they are similar, but not to what. Does it mean similar to those in S2?

Sea ice and temperature patterns are similar to the anomalies found in the S2 experiments. Changed to

"In the S1 experiments temperature and sea ice anomaly patterns are weaker but similar to S2 are found and S1_CHEM is characterized by an amplified temperature reduction as well (not shown)."

L12 add comma after "sea ice formation"

done

L15 Here I wonder how relevant it is for the sea ice increase the advective contribution from a stronger AMOC.

Unfortunately, we do not understand this comment.

L17 Replace everywhere in the text Nordic Sea for Nordic Seas, as it stands for Greenland, Norwegian, Iceland seas, and sometime also the Barents Sea.

Done.

L23 please, rephrase ". . . but the significance is reduced"

Rewritten to: "The anomalies in the S1 experiments are similar, but the significance of the differences to the CTRL is lower."

L24 add comma before while

done.

L28 This sentence is probably too long. It could be divided into two. L30 please, clarify or rephrase "in other parts of the North Atlantic"

Rewritten to: "In S2\_CHEM, however, a reduction of the density is found near the entrance of the Labrador Sea. This causes a reduction of the deep water formation in this area during the first half of the SRR, which is partially compensated by positive anomalies in the eastern North Atlantic (Fig. 3a)."

L30 remove comma after period L33 remove comma after convection L35 rephrase "Similar to the Nordic Seas" (for example, "As in the Nordic Seas,")

Applied as suggested.

**Page 7**

L6 add comma after forcing L7 Split the sentence into two. "in comparison to S2_NOCHEM. Similar differences . . . "

Applied as suggested.

L11 This statement might need a citation

We have clarified the statement and we have added a reference:

"Chemistry-climate interactions are the most pronounced in the stratosphere (e.g., Dietmüller et al., 2014)."

L17 add comma after forcing L21 add comma before a reduction L23 add comma after Furthermore

Done.

L25 It is interesting to notice that changes in the polar vortex do not seem to go linearly with the reduction in the solar forcing. One should not expect linearity in the response, of course, but it is interesting in any case.

Indeed. We mention this explicitly in the revised manuscript: "These responses highlight the non-linear relationship between the solar forcing and the atmospheric dynamics."

**Page 8**

L9 add comma after response; change phenomena for phenomenon L10 add comma after AO index

Done.

L12-14 I do not necessarily agree with the authors on some of the interpretations they make from Figure 6 regarding the AO index, which are in these lines exposed. For example, changes in the S1 experiments are mostly nonsignificant, and, although in CHEM there is a shift toward more negative values, in NOCHEM the change is more like a broadening of the distribution, rather than a change to more negative phases. Also, it should be stated here that the AO index in S2_NOCHEM features a first half of mostly negative values, followed by a positive trend towards more positive. This might even be investigated further, as an extra.

We agree, that the significance of the anomalies is weak. Therefore, we have included the boxplot in Figure 6, which shows the statistics of the AO index, averaged over the 30 year SRR period. This supports our findings with a shift towards more negative AO values with reduced solar forcing and the clear difference between experiment with and without interactive chemistry. A widening of the distribution is not visible in the boxplots.

The higher years with negative AO in the first half of the solar minimum and the shift to neutral conditions in the second half is an interesting feature in S2_NOCHEM, which we mention now in the revised manuscript:

"In particular, negative AO phase tend to occur more often in the first half of the SRR period, while neutral conditions dominate in the second half."

L16 affects

done.

L25 Here a statement connecting changes in temperature and salinity with those in density might be help connect ideas.

Rephrased to: "Since the density of the water decreases with increasing temperature and decreasing salinity, all these changes lead to a pronounced reduction of the mixed layer depth (Fig.f7f)."

L27 Could you explain shortly or cite in the literature why this instantaneous AMOC response to the AO? Is it due to wind forcing? If it were due to heat-driven changes in the convection, as those found during positive or negative phases of the NAO, I would assume some delay in the response of the AMOC

Is it a combination of wind stress and heat flux changes. Delworth and Zeng (2016) performed sensitivity experiments where the forced an ocean model by artificial atmospheric forcing. In one of their experiments they instantaneously switched the atmospheric forcing to an NAO positive state. Their results show that after about 5-7 years the AMOC responds to this forcing with strengthening of the circulation (compare Fig. 3 in Delworth and Zeng). This shift of a few years agrees with our results, although an exact timing is difficult to estimate from our results. In our Fig. 6 we see that it takes a few years before the AO shift towards a predominant negative phase in S2_CHEM (about year 10 of the simulations). Differences in the AMOC, however, emerge around the year 20 (Fig. 1c), so about 10 years after the AO shift.

In the revised manuscript we will improve the discussion of this effect.

L33 Add comma after As a consequence,

done.

L34-35 Isn't it a reduction in the density? Otherwise, one should not expect a reduction in the convection, but an intensification

Right, thank you. This is corrected in the revised manuscript.

**Page 9**

Conclusions: I'd call this section Conclusions and Discussion. L6 please, remove comma after chemistry L12 the sentence about the projected future weakening of the AMOC should be connected with the next paragraph

Applied as suggested.

L15 It would be recommendable to compare the magnitude of the projected minimum with that of those implemented in this study, as well as its duration. If the magnitude of this future minimum were much smaller, we might then expect negligible changes in the AMOC strength.

The magnitude of the future solar minimum is at least as uncertain as the magnitude of past solar minima. For the past, proxy based solar forcing reconstructions indicate TSI amplitudes between 6 $Wm^{-2}$ and below 1 $Wm^{-2}$ (for the TSI difference between the Maunder Minimum and present day). While a reduction of 20 $Wm^{-2}$ (S2 experiments) is clearly out of this range the S1 experiments (-3.5$Wm^{-2}$) are not completely unrealistic.

We have added a sentence on the uncertainty of future TSI change to the revised manuscript:

"Several studies suggest that the sun may enter a grand solar minimum within the next 100 years (Lockwood et al.,2009; Steinhilber and Beer, 2013; Roth and Joos, 2013), although the amplitude of the TSI change is associated with large uncertainties."

L20 please, rephrase. For example, adding after effect "when atmospheric chemistry is taken into account"

Rephrased as suggested.

L25 Many of the elements?

Rephrased to "Parts of the dynamic effect…"

L26 on various time scales. Also, it would be recommendable to indicate which scales in particular the authors refer here

This will be considered in the revised manuscript.

L25-30 In this paragraph, three different verb tenses are used to talk about results from previous studies. I suggest using only one, maybe past simple?

We will rephrase this paragraph is suggested.

L31 remove comma after for the first time

Done.

**Page 10**

L2 when chemistry-climate interactions. . . this, I think, is already indicate at the beginning of the sentence L4 remove comma after GHGs L7 add comma after In this case,

Done.

**FIGURES** Would it be recommendable to add some of the Supplementary Figures to the main text? In particular those that are most referred in the text. There are indeed more Supplementary Figures that main ones.

We will consider this when revising the manuscript.

Fig. 1 Please, clarify whether the Student's t-test done after or before smoothing? The gray vertical lines indicating the SRR period are black

The Student's is performed using the annual mean values, therefore before the smoothing was applied. This is stated in the caption: "Thick dots denote significant differences in the (un-smoothed) annual mean values between the SRR ensemble and the control ensemble (Student's t-test, $p \leq 0.05$)."

We replaced the figures with new versions using a lighter grey colour.

Fig. 2. Please, clarify why climatologies in panels e and g, and in f and h are different, if they derive from the same control simulation, CHEM and NOCHEM respectively?

We assume the reviewer is referring to Fig 3. The ctrl contour lines are different between e/g and f/h because the ctrl contours are calculated over the same period that was used to calculate the anomalies. Anomalies are expressed for the first and second 15 years of the solar minimum.

Figs. 5 and 6 Gray vertical lines are again black

see above.

Fig. 7 Readjust text to match the panels

Thank you, the caption has been corrected.

Fig. 8 Could you please increase the font size of the smallest text?

Font size has been increased.

Fig. S4. What are the shading and contours respectively?

We have included this information to the revised caption: "Contours and shadings from --8 to 8 m/s (contour step 1 m/s)."

**References**:

- Anet, J. G., Muthers, S., Rozanov, E., Raible, C. C., Peter, T., Stenke, A., Shapiro, A. I., Beer, J., Steinhilber, F., Brönnimann, S., Arfeuille, F., Brugnara, Y., and Schmutz, W.: Forcing of stratospheric chemistry and dynamics during the Dalton Minimum, Atmos. Chem. Phys., 13, 10951-10967, doi:10.5194/acp-13-10951-2013, 2013.

- Anet, J. G., S. Muthers, E. V. Rozanov, C. C. Raible, A. Stenke, A. I. Shapiro, S. Brönnimann, et al. 2014. "Impact of Solar versus Volcanic Activity Variations on Tropospheric Temperatures and Precipitation during the Dalton Minimum." Climate of the Past 10: 921–938. doi:10.5194/cp-10-921-2014. http://www.clim-past.net/10/921/2014/.

- Muthers, S., J. G. Anet, a. Stenke, C. C. Raible, E. Rozanov, S. Brönnimann, T. Peter, et al. 2014. "The Coupled Atmosphere–chemistry–ocean Model SOCOL-MPIOM." Geoscientific Model Development 7: 2157–2179. doi:10.5194/gmd-7-2157-2014. http://www.geosci-model-dev.net/7/2157/2014/.

---

## Author Comment (AC4) · 6 Jul 2016

We would like to thank referee 1 for his/her constructive and detailed review.

——————- General comments ——————–

This paper is presenting different sets of simulations that evaluate the impact of a decrease in Total Solar Irradiance (TSI) over three decades, with a specific attention to the AMOC. It is focusing on the impact chemical changes induced by such a decrease, through comparison of a model not including this process, and another one including it. In both models, the decrease of TSI leads to an AMOC strengthening in the decades following the onset of the decreased TSI. The authors argue that this strengthening is larger when the chemical processes are not accounted for. They attribute such an effect to the impact of stratospheric chemistry has on the AO response to TSI decrease. Indeed, TSI decrease may lead a negative NAO due to larger cooling in the stratosphere associated with ozone depletion, which when reaching the surface may affect air-sea fluxes and wind stress, decreasing in particular salinity, which may diminish salinity in the ocean convection sites, limiting AMOC enhancement.

As the former summary shows it, the amount of results shown in this paper is very significant. The topic is also of large interest, since the climatic impact associated with AMOC is well known as well as its good predictability a few decades ahead, and the TSI is also potentially largely predictable and is believed to decrease substantially in the coming decades. The impact of chemistry in the stratosphere was believed to potentially impact the AMOC response to TSI (e.g. Ottera et al. 2011), and this is the first study I see that tackle this potentially important process.

The paper is generally correctly presented, even though I have a large number of comments to clarify and better present the results. My main concerns are that:

1. the main effect analysed (i.e. the impact of chemistry on AMOC response to TSI decrease) is very small and maybe hardly significant;

We do not agree with this comment of reviewer 4. In Figure 1 c the differences in the AMOC behaviour between S2_CHEM and S2_NOCHEM is very clear and highly significant. Furthermore, significant (but weaker) AMOC differences are found between S1_CHEM and S1_NOCHEM, which confirms the results found in the S2 experiments.

2. the demonstrations are sometimes too rapid;

We clarified several steps of our analysis in the revised manuscript.

3. the amount of nice results is maybe too large, which may request to separate the analysis into two papers, i.e. two parts of the main analysis. The first dedicated to a better understanding of AO/NAO response, which is already largely depicted in the present paper, and constitute a very important results, even if not new. The second one will be dedicated to the analysed of the AMOC, which deserves a few more analysis, especially since it is the main topic of the present paper, but only have a few figures that are directly analysing the process involved in the presented changes.

We do not think that the results should be separated into two papers. The response of the AO/NAO to the stratospheric changes has been reported in numerous previous studies. The response of the AO to either solar forcing or the AO/NAO has also been reported previously. The two topics would therefore confirm previous results but would not present novel results.

The novelty of our study is that we can show that all these (previously reported) processes modulate the response of the AMOC to a reduction in the solar forcing and that, furthermore, interactive chemistry has a strong effect for the response. We are convinced this should be presented in a single paper.

Concerning the impact of the AMOC, I'm not entirely sure that the effect of chemistry leads to significant results. The ensemble mean of the simulation seems a bit different, but no error bar, nor statistical test are applied to confirm the supposed impact. Generally speaking, the differences between the two sets should be more systematically highlighted as in Fig. 4 (right panels), which is not the case everywhere, as well as the error bar associated with ensemble spread. Since this is the main result highlighted in the paper, this should be proven with more statistical confidence, or the main message of the paper should be modified.

We have added several figures showing the differences and significance tests for the differences to the revised manuscript.

For all these reasons, although I found the set of experiments very interesting and potentially improving our understanding of climate dynamics in response to solar forcing, I found the take-home message and general descriptions of the results and logical connections sometimes a bit rapid. I therefore consider that the manuscript need major revisions before to be published, and I will advise the authors to consider the possibility of splitting their results into two parts (and two papers) in order to describe properly the main results and mechanisms discussed.

We rewrote parts of the manuscript to improve the presentation of our results. However, we do think that splitting the paper into two parts is appropriate (see above).

——————- Specific comments: ——————-

- P. 1, l. 20: "is responsible for the temperature conditions in western Europe": there is a lot of debate on this specific topic: cf. Seager et al. (2002). The AMOC does not have only an impact on western Europe and cannot explain the whole climate of this region

We fully agree. We changed this sentence to:

"The surface branch of the AMOC transports heat from the Southern Hemisphere and the tropics towards the North, is closely connected to the Atlantic Multidecadal Oscillation, and contributes to the temperate climatic conditions in western Europe (Knight et al 2006)."

- P. 1, l. 22: "Meehl et al. 2009b": 2009a should come first.

This is corrected in the revised manuscript.

- P. 1, l. 23: add "in the past" after climatic changes"

done.

- P. 2: l. 13: "eruptions have been found to intensify the AMOC on decadal time scales": this is not just a question of intensification, but rather of variability excitation cf. Swingedouw et al. (2015)

We modified the manuscript accordingly:

"Moreover, volcanic eruptions may excite the variability of the AMOC (Swingedouw et al., 2015)."

- P. 4, l. 30: "monthly mean": This is a surprising choice. By doing so you include large part of so called Ekman wind-driven variability. Have you tried to remove this component, or to consider annual mean to limit its influence.

We have corrected this in the revised manuscript. We use the annual mean of the streamfunction to calculate the AMOC index.

- P.5, l. 3: what are the spread or error bar associated to the value given (since we are here considered ensemble of simulations.

The temperature difference between S2_CHEM and S2_NOCHEM is not a major point of our study. Therefore we did not present a detailed assessment of the differences between the two experiments. We have added two ensemble spread in the Table below.

| Experiment | ΔT (K) | standard deviation (K) |
|---|---|---|
| S2_CHEM | -1.0 | 0.04 |
| S2_NOCHEM | -0.9 | 0.07 |
| S1_CHEM | -0.1 | 0.10 |
| S1_NOCHEM | -0.1 | 0.07 |

The Student's t test suggest that the differences between the S2 experiments are highly significant (p=0.0009984929), while the differences between CHEM and NOCHEM are not significant in the case of the S1 forcing.

- P. 5, l. 8: can you be more specific on the reference that gave the climate sensitivity of the model and the computation you have made. When you gave numbers, you have to be more specific on the way you compute them.

The climate sensitivity of SOCOL-MPIOM and the experiment are described in Muthers et al. (2014). A reference is given in the model description but we will add the reference to this point as well.

- P. 5, l. 19-20: why is outgoing longwave increasing when water vapour increases. Please clarify the process at play here.

Stratospheric water vapour decreases in the SSR experiments, therefore OLR increases. The ordering of the sentences in the submitted manuscript was a bit misleading. We changed this to:

"In S2_NOCHEM, the largest anomalies (-15% ) are found in the tropical upper troposphere, but stratospheric reductions exceed -10%  almost everywhere (Fig. S1c). In S2_CHEM, the stratospheric

reductions in water vapour are more pronounced (up to -35% ), due to the effect of the solar forcing on the oxidation of methane, the most important in-situ source of stratospheric water vapour (Fig. S1b). Due to the greenhouse effect of ozone and water vapour, the outgoing long-wave flux increases more in CHEM than in the NOCHEM and leads to an additional cooling of the troposphere. The positive water vapour anomalies found in the uppermost model levels in the CHEM experiments (Fig. S1b and e) are related to the reduced UV photolysis of the water vapour molecules."

- P. 6, l. 1: why don't you look at the NAO rather than the AO, since you are looking at the North Atlantic region. The two are usually very much linked, but can you confirm this in your model?

For the analysis of the stratosphere troposphere interactions the AO index is the more appropriate parameter. The AMOC index, however, is stronger influenced by the NAO. In our study, the influence of the stratosphere on the AMOC is analysed and therefore, we have to decide for one of the two indices to draw a consistent picture, from the stratosphere down to the ocean.

We argue, however, that the decision for one index or the other does not affect the conclusions of our study. Both indices are closely related in SOCOL-MPIOM. For CHEM_CTRL we find correlation coefficient of about 0.71 (0.68 in NOCHEM_CTRL.) For comparison we also performed our analysis with the NAO index (defined by the pressure difference between Iceland and the Azores). The results are given below and do not differ strongly between the results for the AO index.

[Figure]

*R 7: Similar to Fig 6 a/b of the submitted manuscript, but for the ensemble mean winter (Nov. – Mar.) NAO index. The NAO is defined by the sea level pressure difference between Iceland and the Azores. Blue lines correspond to the NOCHEM experiments, orange lines to the CHEM experiments. Solid lines resemble the S2 and dashed lines the S1 experiments. Dots indicate winters with significant differences to the CTRL ensemble (Student t-test p ≤ 0.05).*

[Figure]

*R 8: Similar to Fig 6 d/e of the submitted manuscript, but for the ensemble mean winter (Nov. – Mar.) NAO index. Blue boxplots correspond to the NOCHEM experiments, orange boxplots to the CHEM experiments.*

[Figure]

*R 9: Similar to Fig 7 of the submitted manuscript, but for the ensemble mean winter (Nov. – Mar.) NAO index.*

- P. 6, l. 9: "the sea ice differences": when? Try to be very precise on what you are talking about

We are referring here to the sea ice differences shown in Fig 2. which are averaged over the 30yr solar minimum period. We clarified this in the manuscript:

"The Arctic sea ice differences between CHEM and NOCHEM, which emerge in the last 20 years of the reduction period, and are related to the weaker AMOC in the CHEM experiments and the reduced heat transport into the Arctic."

- P. 6, l. 9: "therefore": the logical connection is not very clear to me, please clarify it.

See our answer to the previous comment (therefore has been removed).

- P. 6, l. 17: "Nordic Sea". I am usually seeing "Nordic Seas", since it is a few seas that you are dealing with (Greenland, Iceland, Norway). The same elsewhere in the ms.

This has been corrected.

- P. 7, l. 8-9: I'm not convinced the anomalies between CHEM and NOCHEM are significant for the AMOC. Please provide appropriate statistics.

We are refereeing here to the differences in the AMOC index between CHEM and NOCHEM. A reference to Figure 1 has been added to the revised manuscript.

The differences between the AMOC in S2_CHEM and S2_NOCHEM are significant (Students t-test, $p < 0.05$) for the years 27, 28, 30, and 32, which corresponds to the end of the solar reduction period. In the revised manuscript we highlight years with significant differences between the CHEM and NOCHEM experiments in Fig. 1.

The significant differences between S2_CHEM and S2_NOCHEM are furthermore obvious from the latitude-depth cross section of the meridional streamfunction in the Atlantic. This figure is included as Fig. S2 in the revised supporting material. In the second half of the solar reduction period the differences are significant in a large region between about 800m to 3000m depth from the Southern hemisphere to 50°N. Between 30°N and 45°N the significant differences reach all the way up to the surface.

- P. 7, l. 11: can you provide a reference or an explanation to support this claim?

We have clarified the statement and we have added a reference:

"Chemistry-climate interactions are the most pronounced in the stratosphere (e.g., Dietmüller et al., 2014)."

- P. 7, l. 13: "28K" is this concerning only a grid points?

The term "up to 28 K" described the maximum temperature difference, which is found in one or a few grid points-

- P. 7, l. 20: "-43%": when? Over the 30-year period?

Yes, -43% when averaged over the 30 year period. We clarified this in the manuscript:

"a reduction of -43% is found in S2_CHEM during the winter season (Nov. to Mar.) when averaged over the SRR period."

- P. 7, l. 23: what is your definition for the "duration of the winter period"?

The winter period starts with the first day with a westerly daily mean zonal mean wind component at 60N and 10hPa after 1. October and ends with the first day with easterly wind after 1. April.

We clarified this in the manuscript:

"Furthermore, the duration of the winter period with predominant westerly wind is reduced in S2_CHEM by −30% and in S2_NOCHEM by −5% respectively, when defining the start of the winter

period by the day with the first occurrence of a westerly daily mean zonal mean wind component at 60N and 10hPa after September and the end by the first day with easterly winds after March."

- P. 7, l. 24-25: a series of number are given, with very poor definition. Please clarify.

See above.

- P. 8, l. 3: "downward coupling": can you define this?

We changed this to "downward propagation".

- P. 8, l. 21: "freshwater flux": from which component? Precipitation? Evaporation? Sea ice?

We are referring to the total freshwater flux from all three processes. We clarified this in the revised manuscript.

- P. 8, l. 23: "export of saline water from the Nordic Sea by EGC". The EGC is a very fresh and cold current, so it is not exporting saline water! Do you mean the weakening of this current is increasing the salinity?

Thank you. We will give a detailed answer to this in the revised manuscript.

Please clarify.

- P. 8, l. 28: "instantaneously": thus, this is likely not related to convection but rather to wind-driven changes. Can you comment on that?

We will rephrase this paragraph. The word "instantaneously" may not be appropriate in this case.

- P. 8, l. 31: "weaker intensification": significant? At which level? (please account for autocorrelation when computing degrees of freedom, since the AMOC has very low variability.

At this point we summarize the previous results. The weaker intensification is shown in Fig. 1c. Dots in Fig. 1 represent year, where the SSR ensemble (e.g., S2_CHEM, 10 simulations) differs significantly from the control ensemble (e.g., CTRL_CHEM, 10 simulations). We therefore do a comparison of two data sets with 10 values each against each other. There is no autocorrelation, since we do not include any temporal information and the 10 experiments can be considered to be independent.

- P. 9, l. 11: "is also one of the": not really, since in projections, this is the longwave radiation that is mainly affected rather than the solar radiation changes.

Our description was a bit misleading. We clarified this in the revised manuscript:

"This response of the overturning to solar radiation changes has been identified in earlier studies (Cubasch et al., 1997; Latif et al., 2009; Otterå et al., 2010; Swingedouw et al., 2011). Related to increasing global greenhouse gas concentrations and associated surface warming, it is also one of the dominant mechanisms for the projected future weakening of the AMOC (Stocker and Schmittner, 1997; Manabe and Stouffer, 1999; Mikolajewicz and Voss, 2000; Gregory et al., 2005; Stocker et al., 2013)."

- P. 9, l. 20-24: while the impact on the AO is very large, the impact on the AMOC is very weak, why is that? Is it coherent with small effect of AO on AMOC in control? What is the regression value of the AO on the AMOC in this model? Lohman et al. (2009) can be an interesting references concerning long term of a positive NAO on North Atlantic.

The response of the AMOC is a combination of several factors. The AMOC responds to the temperature changes, which cause an intensification of the overturning. In the CHEM experiments the AMOC is furthermore affected by the AO, with the negative AO phase leading to a weakening. Therefore, the response of the AMOC to the AO is already counteracted by the effect of the temperature changes.

We discuss the study of Lohmann et al. in the revised manuscript:

"The influence of the dynamic effect on the AMOC may furthermore depend on the length of the solar reduction period. Lohmann et al. (2009) found a gradual weakening of the subpolar gyre response with time in ocean model simulations forced with a persistent negative phase of the NAO. Additionally, the response of the AMOC may be non-linear and an increase of the solar forcing may change the dynamic effect (Lohmann et al., 2009)."

- P. 9, l. 32: "importance": I think this is a strong statement for a very weak effect in the end. . .

We do not agree. Between S2_CHEM and S2_NOCHEM we find an AMOC difference of about 1 Sv, which can only be attributed to the interactive chemistry. This difference cannot be considered as weak. Moreover, the weaker forced S1 experiments also reveal a clear difference in the AMOC intensities between S1_CHEM and S2_NOCHEM.

- P. 10, l. 1: add "slightly" after "may"

see above, we do not think that the response is weak, therefore a "slightly" would reduce the importance of our study.

- Fig. 1: please compute a statistical test for differences between CHEM and NOCHEM anomalies.

See above.

- Figs 2,3, 7: please compute the difference CHEM-NOCHEM as in Fig. 4

For the revised manuscript we will produce a plot of the differences between CHEM and NOCHEM for these figures. If additional informations can be drawn from these figures we will include them in the revised manuscript, otherwise we will show the in our final answers to the reviewer.

- Fig. 7: This is a key figure when trying to understand what is going on for the AMOC, which should be the heart of the paper, given the title. Why is the projection so different than in 3? We want to see what is going in the whole Nordic Seas, including Fram Strait. What about circulation changes (barotropic stream function for instance)? Wind stress? Density? Thermal and salinity component of density? The demonstration of the processes affecting the North Atlantic should be more depicted.

We will address these questions when revising the manscript.

Figure S2 in this regard is interesting and should come in the main ms., but what is missing on this figure is an indication of the time frame. When are the changes occurring. Each point corresponds to a year from what I understand (with a smoothing of 15 years). Thus the anomalies are firstly thermally driven and then salinity driven. Why is there such a 10-year lag? (which is not clear from Fig. 8 where no time scale is shown).

Thank you, we will think about including this figures to the main manuscript.

On the time-lag question: Delworth and Zeng (2016) performed sensitivity experiments where the forced an ocean model by artificial atmospheric forcing. In one of their experiments they instantaneously switched the atmospheric forcing to an NAO positive state. Their results show that after about 5-7 years the AMOC responds to this forcing with strengthening of the circulation (compare Fig. 3 in Delworth and Zeng). This shift of a few years agrees with our results, although an exact timing is difficult to estimate from our results. In our Fig. 6 we see that it takes a few years before the AO shift towards a predominant negative phase in S2_CHEM (about year 10 of the simulations). Differences in the AMOC, however, emerge around the year 20 (Fig. 1c), so about 10 years after the AO shift.

We will discuss our finding in the light of the study by : Delworth and Zeng (2016) I the revised manuscript.

—————- Bibliography: —————-

- Lohmann K, H Drange, M Bentsen (2009) Response of the North Atlantic subpolar gyre to persistent North Atlantic oscillation like forcing. Climate dynamics 32 (2) pp 273-285

- Seager R, DS Battisti, J Yin, N Gordon, N Naik, AC Clement and MA Cane (2002) Is the Gulf Stream responsible for Europe's mild winters? QJRMS 128 (5), pp. 2563-2586

- Swingedouw D, P Ortega, J Mignot, E Guilyardi, V Masson-Delmotte, PG Butler and M Khodri (2015) Bidecadal North Atlantic ocean circulation variability controlled by timing of volcanic eruptions.Nature Communications 6, pages: 6545

---

## Author Response (AR1)

**We would like to thank referee 1 for his/her constructive and detailed review.**

This is a mostly well written manuscript with interesting new results identifying a stratospheric mechanism impacting the Atlantic Meridional Oscillation. Therefore, it is potentially suitable for publication in the Earth System Dynamics journal. I have, however, a few concerns which I would like the authors to address before I can recommend the publication.

**My major concerns are:**

1. What would be the impact of aerosols? Your model does not include aerosol interactions, you just simply reduce the solar radiation. This seems a critical simplification to me. You should at least discuss how aerosol interactions would modify the AMOC response if taken into account in your model.

We agree, that the response of the coupled atmosphere-chemistry-ocean model to stratospheric aerosols is the next follow up question, which should be addressed. In this study, however, we focus on response of the system to a direct reduction in the solar energy input (i.e. total solar irradiance). A reduction of the TSI takes place during grand solar minima (e.g., Dalton Minimum) or in the case of solar radiation management techniques taking place in space (e.g., reduction of the TSI by mirrors in space). The model includes also aerosol interactions and indeed a number of modelling studies on the response to stratospheric aerosols from volcanic eruption have been performed earlier (e.g., Anet et al, 2013, Muthers et al. 2014 and 2015).

A comparison between both approaches, a reduction of the TSI in space and through stratospheric aerosols is, however, highly relevant. We therefore discuss possible effects of radiation management by stratospheric aerosols at the end of the submitted manuscript.

"The dynamical effect is expected to change, however, when the solar radiation is reduced in the Earth's atmosphere, for instance, by stratospheric sulphate aerosols. In this case a strengthening of the NH polar vortex and a positive phase of the AO may develop, analogous to the response to strong tropical volcanic eruptions (Graf et al., 1993; Kodera, 1994; Stenchikov et al., 2002; Muthers et al., 2014a, 2015). This effect of the positive AO phase may, in turn, lead to an intensification of the AMOC. Future studies shall address the influence of stratospheric sulphate geoengineering on the AMOC and the possible role of chemistry-climate interactions."

2. I think in reality the salt rejection from the sea-ice growth is rather small and mainly occurs north from the regions of deep convection. Therefore, it has only a minor importance to the deep convection and the AMOC compared to the heat loss and possibly the net precipitation (precipitation minus evaporation) at the ocean surface. At the moment, the reader is led to understand that the salt rejection is at least as important as the heat loss. The increase of the sea-surface salinity could also be due to a decreased net precipitation related to changing storm tracks, for example. To better support the salt rejection argument, you need to quantify the salt rejection to the surface density and compare it to other factors. Can you check the ocean surface fluxes from your model output and their relation to the T and S, not only density, anomalies? How realistic these modelled fluxes then

are, depend on your model skill and are related to your model configuration, such as the sea-ice salinity scheme.

Figure 4 (formerly S2) of the revised manuscript shows, that salinity changes contribute substantially to the density changes in both deep water formation regions, in particular in the second half of the reduction period. Figure R1 (below), furthermore shows the spatial pattern of the anomalies. Both, sea-ice growth and precipitation changes contribute to this increase of the salinity. So far, we cannot answer, which process is more important for the increasing salinity, therefore we mention both processes in the revised manuscript.

Figure 4 furthermore shows, that heat loss is clearly more important in the first half of the reduction period. We therefore, carefully rewrote the results section of our manuscript and took care not to overemphasize the importance of sea ice growth.

*R* 1: Upper ocean (0-100m) salinity anomalies in S2\_CHEM (left) and S2\_NOCHEM (right) for the first (top), second (middle), and last (bottom) 10 years of the solar reduction period. Ensemble mean anomalies are calculated relative to the corresponding control ensemble mean. Dots denote significant differences (p<0.05).

3. I have a problem when you treat the AO and NAO identically. Although the AO and NAO correlate, they are not identical, not even from the AMOC perspective. I agree that the AO behaves largely like the NAO in winter. If, instead of the AO, you based your analysis on the NAO, how would the results look like? What would be their significance after taking into account the possible year-to-year autocorrelation?

**See in comments below.**

**Minor comments:**

- Page 1, line 16. I would rather say that 'surface currents transport water into the northern North Atlantic' rather than to 'Northern high latitudes' which sounds more like to the Arctic Ocean.

**We changed 'Northern high latitudes' to 'North Atlantic'.**

- Page 2, line 7. I don't think the AO is the hemispheric equivalent of the NAO. The NAO is a regional index and correlates with the AO, but their definitions differ substantially.

**We have deleted the phrasing 'hemispheric equivalent'**

- Page 2, lines 18. '... by increasing SSTs and enhancing freshwater input ...'

Rewritten to: "An increase in the solar forcing has been found to weaken the AMOC by increasing SSTs and enhancing freshwater input (Cubasch et al., 1997; Latif et al., 2009; Otterå et al., 2010; Swingedouw et al., 2011)"

- Page 2, line 32. As you focus on the AO in this paper, would be clearer not to talk about the NAO, but the AO, after Page 2, line 7.

We agree and focus on the AO in the revised manuscript. Note, that the results are very similar, when the analysis is performed using the NAO index. Some of the relevant literature, however, focuses on the NAO, therefore we cannot completely remove the NAO from the manuscript.

- Page 3, line 19. '... uses temperature data ...'

**Done.**

- Page 4. line 3. You provide very little details on the model configuration. For example, what was the time step you used? How about the sea-ice salinity, was it constant? Or what sea-ice thermodynamics scheme was deployed? This information is important to assess how realistically the sea-ice salt rejection was modelled.

The model used is a configuration of the widely used ECHAM5-MPIOM (COSMOS model), which has been applied in various modelling studies and the IPCC AR4. The only difference between our version and the COSMOS version is the coupled atmospheric-chemistry module and this configuration is described in great detail in Muthers et al. (2014b).

The requested information has been included in the model description of the revised manuscript:

• Sea ice thermodynamics: "Sea ice dynamics are based on the viscous-plastic rheology formulated by Hibler (1979)."

- "A constant sea ice salinity of 5 psu is assumed."
- Time-step: "The time-step of the atmospheric component is 15 minutes, with the full radiation and the chemical computations updates are performed every 2 hours." Ocean: "The time-step of the oceanic component is 2 hours and 24 minutes."

- Page 4, line 20. You should mention here how long model simulations continued after the 30 year SSR period.

Rewritten to: "The reduction of the solar forcing is switched on in year 5 of a simulation and lasts for 30 years when it is switched off and the simulation is continued for 25 years."

- Page 4, line 27. Explain the acronym TSI.

The acronym TSI is now defined at its first occurrence.

- Page 4, line 32. Explain more in detail how the AO index was calculated and provide references. For example, a common way to calculate the AO is based on the PC1 of 1000 mb p

---

## Author Response (AR2)

Editorial side notes:

1) Figure S4 caption:
- "Oktober" should read "October".

We have corrected this in the revised manuscript.

2) Figure S7:
Red-green colour combination: if you find it easily feasible and appropriate, you might consider readjusting the colour scheme to avoid coexistence between red and green. The excellent colour scheme consistently used in the other figures (e.g. S5, S6) could yield a good colour combination involving e.g. blue, dark yellow and red.

Thanks for this comment. We moved to another (colour blind friendly) colour scheme.

Anonymous Referee #3
* * *
* accepted subject to minor revisions

The authors have shown convincingly that atmospheric chemistry can play a potentially important role in setting North Atlantic Ocean variability, particularly during periods of low solar activity. This is relevant for better understanding not only past but future climate changes; therefore, the manuscript should definitely be published.

With respect to the previous version, the manuscript has gained clarity and feels sounder. It is well written and Figures support well the discussion of the results. The authors have also addressed the raised issue about the Introduction, and now we all can clearly and at once understand the importance and novelty of this contribution.

I only have some small editorial and specific comments that should be considered before publication:

L7 and L11: These two lines somehow contradict each other: "In simulations with chemistry-climate interactions a second, dynamical effect on the AMOC is identified" vs. "The dynamic mechanism is present in both ensemble[s]". Please, clarify.

We rewrote parts of the abstract to in the revised manuscript:

"... Furthermore, a second, dynamical effect on the AMOC is
identified driven by the stratospheric cooling in response to the reduced solar forcing. The cooling is strongest in the tropics and leads to a weakening of the Northern polar vortex. By stratosphere-troposphere interactions, the stratospheric circulation anomalies induce a negative phase of the Arctic Oscillation in the troposphere which is found to weaken the AMOC through

wind stress and heat flux anomalies in the North Atlantic. The dynamic mechanism is present in both ensemble experiments. In the experiment with interactive chemistry, however, it is strongly amplified by stratospheric ozone changes…"

L8 "atmosphere-ocean-chemistry climate model"

thank you.

L20-onwards: Again, I think "temperature reduction" could be rephrased by "cooling" most of the times throughout the text.

We replaced 'temperature reduction' by 'cooling' in several places throughout the manuscript.

L29 add "(Fig. 2)" after "than in S2_NOCHEM"

Done.

L17 please, rephrase: "hardly significant" (mostly non-significant, maybe?)

Rephrased to: "A further intensification is found for the second and the third decade, but the anomalies between S2_CHEM and S2_NOCHEM show only a weak significance."

L17-18 please clarify "the significance […] is lower" (differences are, again, mostly non-significant?)

Rephrased to: "The anomalies in the S1 experiments are similar, i.e., differences are mostly non-significant."

L21 "larger" than what?

Rephrased to:
"In the North Atlantic the density and mixed layer differences between S2\_CHEM and S2\_NOCHEM are larger than the ones in the Nordic Seas."

L23 the "response is found in S2_CHEM" rather than NOCHEM, right?

Of course, thank you!

Figure 2:

L3: "for the sea ice extent"

Done.

Figure 9:

I still think that the smallest font size is too small and could be enlarged

The font size was enlarged by 1points.

Anonymous Referee #4
* * *
* accepted subject to minor revisions

General comments
* * *
This is my second review of this paper. The authors have correctly answered to my concern and convinced me to keep the paper as it stands (i.e. in one part). I mainly have only a few technical issues.
My only point is that I still think that the amplitude of the impact of the stratospheric chemistry on the AMOC is relatively modest. In particular, it should be clearly stated that it remains lower than the direct effect of radiative impact by TSI decrease.
One could have imagined that the impact of chemistry on the NAO could have simply reversed the impact of solar variations, which is clearly not the case here as seen on Fig. 3. I think this should be clearly stated on Fig. 9 for instance, which is only qualitative as it stands at the moment.

From our results we can not conclude, that the impact of the stratospheric chemistry on the AMOC is always smaller than the direct thermal effect. In case of the S1 experiments (Fig. 1 c, lower panel) an intensification of the AMOC is found in the second half of the reduction period in the S1_NOCHEM experiment, while no intensification is found for S1_CHEM. For this scenarios, the two effects seem to cancel out. For the S2 experiments, however, the thermal response is dominant. Therefore some AMOC intensification is found in both experiments. We have included a paragraph on the differences in the response between S1 and S2 to the discussion section of the revised manuscript.

"For the weaker S1 SRR, both effects cancel out and therefore no
AMOC intensification is found in the experiments with interactive chemistry. In the S2 experiments with stronger forcing, however, the thermal response of the AMOC dominates and the dynamical effect leads only to a reduced intensification of the overturning."

Specific comments:
* * *
- P. 2, l. 4: typo on "external"

Done

- P. 5, l. 13: "NAO is closer related" should be "NAO is more closely related"

Done.

- P. 11, l. 3: "The stratosphere responds very fast" should be "The stratosphere responds very rapidly"

Done.

- P. 11, l. 4: typo "is takes" should be "it takes"

Done.

- Fig. 1: the style chosen to highlight significance is not very well-chosen. Indeed, one cannot distinguish a differences significantly different from control and chemistry, since a big dot will overwrite a small one.

In the revised manuscript, we indicate years with significant differences between CHEM and NOCHEM now by small black stars, which are slightly below the other significant dots.

- Fig. 1: I'm a bit puzzle that between CHEM and NOCHEM is significant for year around 45 (thick orange dots): CHEM and NOCHEM values are extremely close. Can you check your code?

We assume the reviewer is referring to panel b of Figure 1, which shows the global mean 2m temperature. The big dot in the year 45 of the experiment indicates significant differences between CHEM_CTRL and NOCHEM_CTRL.
The ensemble mean and the ensemble standard deviation for this year are given in the table below and yield a p-value of 0.04. The differences are therefore significant.

[revised manuscript text omitted]